

# Use of a Heated Graphite Scrubber as a Means of Reducing Interferences in UV-Absorbance Measurements of Atmospheric Ozone

Andrew A. Turnipseed, Peter C. Andersen, Craig J. Williford, Christine A. Ennis, and John W.

Birks

2B Technologies, Inc., 2100 Central Ave., Boulder, CO 80301
*Correspondence to:* Andrew Turnipseed (andrewt@twobtech.com)

**Abstract:**

A new solid-phase scrubber for use in conventional ozone photometers was investigated as a means of reducing interferences from other UV-absorbing species and water vapor. It was found that when heated to 100-130°C, a tubular graphite scrubber efficiently removed up to 500 ppb ozone and ozone monitors using the heated graphite scrubber were found to be less susceptible to interferences from water vapor, mercury vapor, and aromatic volatile organic hydrocarbons (VOCs) compared to

conventional metal oxide scrubbers. Ambient measurements from a graphite scrubber-equipped photometer and a co-located Federal Equivalent Method (FEM) ozone analyzer showed excellent agreement over 38 days of measurements and indicated no loss in the scrubber's ability to remove ozone when operated at 130°C. The use of a heated graphite scrubber was found to reduce the interference from mercury vapor to ≤ 3% of that obtained using a packed-bed Hopcalite scrubber. For a series of

substituted aromatic compounds (ranging in volatility and absorption cross section at 253.7 nm), the graphite scrubber was observed to consistently exhibit reduced levels of interference, typically by factors of 2.5 to 20 less than with Hopcalite. Conventional solid-phase scrubbers also exhibited complex VOC adsorption and desorption characteristics that were dependent upon the relative humidity (RH), volatility of the VOC, and the available surface area of the scrubber. This complex behavior involving humidity is

avoided by use of a heated graphite scrubber. These results suggest that heated graphite scrubbers could be substituted in most ozone photometers as a means of reducing interferences from other UV-absorbing species found in the atmosphere. This could be particularly important in ozone monitoring for compliance with the U.S. Clean Air Act or for use in monitoring indoor air quality.





## 1  Introduction

Ozone ($O_3$) in the lower atmosphere is produced by a complex set of photochemical reactions involving oxides of nitrogen and volatile organic compounds (VOCs). It is well established that anthropogenic emissions of these precursors lead to the production of elevated levels of ozone (Haagen-Smit and Fox, 1954; Chameides and Walker, 1973). It also is recognized that high levels of ozone are

deleterious to human health (White et al., 1994; Weisel et al., 1995; Burnett et al., 1997; U.S-EPA, 2016) and lead to plant and crop damage (Avnery et al., 2011; Emberson, 2003; Emberson et al., 2009). For these reasons, the Clean Air Act in the U.S. and similar laws in other countries have set limits on ozone concentrations in ambient air. Enforcement of compliance with the U.S. National Ambient Air Quality Standards (NAAQS) requires monitoring of ozone concentrations at hundreds of locations, especially

during summer months when photochemical ozone production is highest.

Currently, compliance monitoring of ozone is done almost exclusively by the method of UV absorbance at 253.7 nm due to the simplicity and reliability of this technique. However, it has been shown that UV photometers suffer from positive interferences from other trace gases that absorb at 253.7 nm such as mercury (Hg) and VOCs (Huntzicker and Johnson, 1979; Grosjean and Harrison, 1985;

Kleindienst et al., 1993; Spicer et al., 2010). In addition, water vapor, which does not absorb at 253.7 nm, is also a significant interference in commercial ozone analyzers (Meyer et al., 1991; Kleindienst et al., 1993) due to its ability to change the transmission of light through the absorbance cell (Wilson and Birks, 2006). Although interferences found in outdoor air normally cause only small errors in ambient ozone measurements (a few ppb at most) due to their low concentrations, these small, unpredictable changes in

response can result in regions being in non-compliance with the EPA ozone standard because high levels of interferences often coincide with high levels of ambient outdoor ozone (Leston et al., 2005; Ollison et al., 2013). The recent downward revision of the standard from 75 to 70 ppb (8-hr average) is expected to increase the number of U.S. counties being out of compliance from 224 to 241 (McCarthy and Lattanzio, 2016), making small ppb-level interferences a more critical issue. Furthermore, there is an increasing

interest in monitoring indoor ozone which are typically 20-80% of that measured outdoors (U.S.-EPA, 2016); however, interfering species such as VOCs and mercury can be at much higher concentrations indoors from attached garages, cooking emissions, unvented heaters, the use of paints, cleaners, and outgassing from building components. This places even more stringent constraints on ozone monitors to reduce biases.


Ambient ozone UV-absorbance measurements are conducted by observing the attenuation of transmission of the mercury emission line at 253.7 nm (near the maximum in the ozone absorption spectrum) through an absorbance cell. Relative light intensities are measured for a direct air sample ($I$) and an air sample where the ozone is removed by some type of ozone scrubber ($I_o$), and the ozone



concentration computed using the Beer-Lambert Law.  Ozone scrubbers are typically composed of

manganese dioxide, charcoal, Hopcalite, metal oxide coated screens, or heated silver wool and play a key

role in the degree to which interferences compromise the ozone readings in UV absorption photometers.

For chemical species that absorb 253.7 nm light, the degree of interference depends on the compound's

absorption cross section, its ambient concentration, and the degree to which the compound is removed by

the scrubber.  If not removed by the scrubber, its concentration remains constant between $I$ and $I_o$

measurements and cancels out in the Beer-Lambert Law. Thus the ideal scrubber would have two

characteristics: (1) it would quantitatively remove ozone; and (2) it would *not* remove other species that

absorb at 253.7 nm or that otherwise modify light transmission to the detector such as water vapor.  The

metal oxide scrubbers used in conventional commercially available ozone photometers do not fully meet

the second criterion, leading to errors in the Beer-Lambert Law calculation of ozone.

75        A further, less obvious problem with UV-absorbing species is that they can desorb from the solid-

phase scrubber at later times, causing a measured negative absorbance (value of $I_o$ is less than $I$).  This

desorption can be caused by temperature or relative humidity changes within the scrubber that alter the

adsorption characteristics of a species and allows it to be released back into the gas phase.  Because UV-

absorbing compounds and mercury are commonly present to some degree in both indoor and outdoor air,

these interferences may be responsible in part for the baseline drift that occurs in photometric ozone

monitors (Birks et al., 2013; Ollison et al., 2013).  Therefore development of a solid phase scrubber that

effectively destroys ozone ($\geq$ 99% destruction), but efficiently passes interfering species is desirable.

Table 1 provides a list of several atmospheric constituents with known 253.7-nm absorption cross

sections along with a selectivity ratio (S), defined as the ratio of the ozone cross section to that of the

interfering compound at 253.7 nm.  Notable among these are VOCs that contain an aromatic group.  The

absorption cross sections of these compounds vary widely, but can be similar to that of ozone, depending

on the types of functional groups attached.  Gaseous mercury is a particularly strong interference because

the electronic energy levels of Hg atoms are resonant with the Hg emission line used in most ozone

monitors.  The relative response of mercury has been estimated at ~1 ppb $O_3$/ppt Hg (Spicer et al., 2010;

Li et al., 2006).  So even a few ppt of mercury vapor can potentially cause substantial biases in ozone

measurements.

In this study we present a means of greatly reducing the interferences of mercury, UV-absorbing

VOCs, and water vapor in ozone measurements by replacing conventional metal oxide internal ozone

scrubbers with a heated graphite scrubber (Birks et al., 2016).  Carbon-based scrubbers have often been

used for removing ozone from air (Shields et al., 1999; Lee and Davidson, 1999).  However, these

scrubbers also effectively remove many other compounds (such as VOCs and water) by adsorption due to

their large surface area (typically 500-3000 $m^2$/g) and thus provide no significant advantage over existing





ozone scrubbers. On the other hand, carbon in the form of graphite has a more ordered planar structure with less surface area (often < 20 m$^2$/g) and fewer sites available for adsorption of possible interfering

compounds. We show that a heated graphite scrubber composed of one or more graphite tubes effectively destroys ozone while efficiently passing most UV-absorbing species and is relatively insensitive to humidity changes. Our results indicate that replacement of conventional scrubbers in commercially available ozone photometers with heated graphite scrubbers can increase their accuracy by reducing potential interferences.


## 2   Experimental

Most experiments were carried out with modified FEM 2B Technologies, Model 202 Single-Beam and Model 205 Dual-Beam Ozone Monitors. Figure 1a shows a diagram of a typical single-beam UV ozone photometer such as the Model 202. Ambient air samples are drawn into the instrument

through an inlet via an air pump. A solenoid valve alternately switches the gas sample directly into the optical detection cell or through an ozone scrubber and then into the detection cell. UV light from a low-pressure mercury lamp UV light passes through the detection cell and is detected by a photodiode containing a built-in interference filter centered at 254 nm. The ozone concentration is calculated using the Beer-Lambert Law,

$$[O_3] = \frac{1}{\sigma L} \ln\left(\frac{I_o}{I}\right) \tag{1}$$

where $\sigma$ is the absorption cross section for ozone (1.15 x 10$^{-17}$ cm$^2$ molecule$^{-1}$), $L$ is the path of the detection cell, $I_o$ is the lamp intensity at the detector in the absence of ozone, and $I$ is the lamp intensity with ozone present. The path length ($L$) is 15 cm for the 2B Technologies Models 202 and 205 Ozone Monitors. The Model 205 analyzer differs in that it is a typical dual-beam instrument containing two

optical cells irradiated by the same lamp, but with each cell equipped with its own photodiode. Two 3-way solenoid valves are switched to alternate passage of ozone-scrubbed and un-scrubbed air through the two cells, providing a simultaneous measurement of $I$ and $I_o$. These analyzers were modified to locate the ozone scrubber outside of the instrument, allowing for easy exchange between the different scrubbers tested. Mixtures of interfering compounds in air were sampled into the ozone analyzers via an overflow

tee where excess air was vented through a second scrubber to the atmosphere.

Graphite scrubbers consisted of different combinations of graphite tubes (¼-in o.d. and 1/8-in i.d., Ohio Carbon Blank, EDM-AF5 graphite) housed in a temperature-controlled aluminum block. Tube lengths were either 7.6 cm (3-in) or 15.2 cm (6-in). Most results were obtained using four 3-in tubes plumbed in parallel as shown in Figure 1b. Using multiple tubes increased both surface area and



residence time while still maintaining high flow conductance. This configuration was chosen because it
      quantitatively destroyed (>99%) ozone in the single-beam Model 202 Ozone Monitor and fit within both
      Model 202 and 205 instruments. The aluminum block was heated with a cartridge heater (Thermal Corp.,
      #CPN20528), whose temperature could be varied up to 160°C and controlled to within ±2°C.  The
      graphite tubes were cleaned before use by heating at 300°C for 12 hours while purging with UHP

nitrogen and sonication for 20 min in dilute Micro-90 glass cleaner followed by rinsing with distilled
      water and drying.  Scrubbers consisting of coarsely ground EDM-AF5 graphite (~ 1-3 mm diameter
      particles) were also tested with similar findings but we report only tube scrubber results because their
      geometric surface area and residence times were more easily characterized.

              Other commercially available scrubbers were included in this study for comparison.  These

include the conventional packed-bed Hopcalite scrubber used in 2B Technologies analyzers, and
      manganese oxide-coated screens from a Thermo Electron Model 49c and a Teledyne-Advanced Pollution
      Instrumentation Model 200E.   We also tested heated silver wool scrubbers similar to scrubbers used by
      Horiba.  It should be noted that although these various scrubbers were used within 2B Technologies
      analyzers, pressure and flow conditions were similar to those in the respective commercial analyzers.

However, small differences in operating conditions of the various commercial analyzers may alter results
      in those instruments compared to those presented here.

              Dry air (< 2% relative humidity (RH)) was produced using a zero air generator (Aadco, Model
      737) or from cylinders of zero grade air.  Ozone/air mixtures and ozone-free air were produced via an
      ozone calibrator – a Model 306 Ozone Calibration Source™ (2B Technologies).  The Model 306

instrument qualifies as an EPA transfer standard for ozone and can produce known mixing ratios of ozone
      via UV photolysis of air in the range 0 and 1 ppm.  Humidity was varied by humidifying the source air for
      the Model 306.  A mixing tee was placed on the source line of the Model 306 with one end pulling
      through a water bubbler and the other through a drying tube (Drierite).  Needle valves on each line
      allowed for independent control over the source line between the dry and humid air.  Humidity was

measured in the overflow just downstream of the ozone monitor via a temperature/humidity sensor
      (Omega, Model HH311).

              Mercury vapor was introduced to the ozone analyzer by using a glass diffusion tube.  A small
      amount of liquid mercury was placed in a glass bulb and vapor was allowed to diffuse through a 20-cm
      long capillary tube (i.d. = 0.5 mm).  At the exit of the capillary tube, a small sweep flow of air (~ 5-15

cc/min) purged the mercury vapor into the main airflow (~ 3 Lpm).  Mercury concentrations were varied
      by immersing the reservoir and capillary tube in an insulated water bath and varying the temperature of
      the bath (10 to 60°C).  Mercury interference measurements were made relative to the 2B packed-bed
      Hopcalite scrubber.





Table 1 summarizes the absorption cross sections at 253.7 nm for a variety of atmospheric species

including compounds tested in this study. Test compounds were chosen because they are either common outdoor (p-xylene) or indoor (phenol) pollutants or for comparison with previous studies (Spicer et al., 2010). Typically, gas mixtures between 5 and 25 ppm were made in pre-conditioned gas cylinders by dilution of the pure compound in UHP nitrogen. These were further diluted with zero grade air via mass flow controllers to produce mixing ratios in the range of 0.01-1.4 ppm. Low volatility (o-nitrophenol)

VOCs were added using diffusion tubes similar to that described for mercury above. Sample purity was assessed using a gas chromatograph with flame ionization and mass spectrometric detection (GC-FID,-MS). The VOC/air mixtures were sampled into ozone analyzers equipped with different scrubbers from an overflow tee. In some experiments, the VOC/air flow was simultaneously sampled into the inlet system of a gas chromatograph equipped with a flame ionization detector (GC-FID). VOC samples were

pre-concentrated via a cooled (-5°C) focusing trap containing Tenax-TA (typically 100-500 $cm^3$ of the VOC/air mixture) and then rapidly heated to inject as described in Greenberg et al. (1994). Both GC systems were calibrated with a NIST-certified primary standard of neohexane in nitrogen or a secondary VOC standard containing isoprene and camphene.

**3    Results and discussions**

As mentioned previously, any species absorbing light at 253.7 nm will be detected if its concentration is changed by the ozone scrubber, and thus an ideal ozone scrubber would remove all ozone but quantitatively pass interferents such as water vapor and UV-absorbing species. The following sections test the graphite scrubber against this ideal, followed by comparative tests with existing

conventional scrubbers for known interferences.

**3.1 Ozone removal efficiency**

It has long been known that ozone reacts with graphite, causing loss of the carbon surface with the assumed production of CO and $CO_2$ as well as other oxidized hydrocarbons (Hennig, 1965; Tracz et

al., 2003; Razumovskii et al., 2007). CO, $CO_2$, and other low volatility hydrocarbons such as acetone and acetaldehyde are readily lost to the gas phase and show negligible absorption at 253.7 nm (Table 1) compared to ozone. A graphite scrubber is not catalytic (i.e., there is actual loss of the scrubber carbon), but it can be estimated that a typical 3-inch long graphite tube would require a continuous 200 ppb ozone exposure for 213 days to lose 1% of its carbon; at 100°C oxidation by ambient $O_2$ is also negligible

(Entegris, 2015).





Initial tests of the graphite scrubber using four parallel 15.2 cm graphite tubes at room temperature showed nearly complete removal of ozone up to 250 ppb in the Model 202; however, the ozone removal efficiency degraded over several hours as the graphite surface became oxidized. Heating the graphite tubes to 100-130°C resulted in complete recovery of the graphite's reactivity toward ozone

but large negative absorbances were observed in the ozone analyzers as the temperature of the graphite scrubber exceeded 145°C. This was due to the emission of various volatile organic hydrocarbons from the graphite tubes as verified by GC-MS. We attribute this to either the thermal breakdown of the organic binder contained in graphite that is added during manufacturing, or to initiation of $O_2$ oxidation, although the latter is not expected to be significant below about 350°C (Entegris, 2015). Below 135°C, emissions

were greatly reduced, and after completing the cleaning process described above, typically resulted in a small negative offset (~ -2 to -5 ppb) that remained constant over time. As in the case of conventional ozone scrubbers, this small offset can be incorporated into the calibration parameters of the ozone analyzer.

We conclude that a tube configuration consisting of four 3.175-mm (1/8-in) i.d., 7.62-cm (3-in)

long parallel graphite tubes (Figure 1b), which easily fit within the existing Model 202 Ozone Monitor enclosure, provides adequate ozone destruction efficiency. This scrubber has an internal volume of 2.4 $cm^3$ and geometric surface area of 30.4 $cm^2$, resulting in a residence time of 0.145 seconds for a typical flow rate of 1 L/min. At a temperature of 100°C and pressure of 1 atm, on average each ozone molecule collides with the surface $1.85 \times 10^4$ times while passing through the scrubber. Thus, assuming rapid

diffusion to the walls, a collisional reaction probability $\gamma$ of ~ $1.7 \times 10^{-3}$ is required to assure >99% ozone destruction. This is similar in magnitude to previously determined probabilities of ozone uptake on fresh carbon soot surfaces (Burkholder et al., 2015). The results discussed in the remainder of this paper were observed with this scrubber design, although the graphite surface area could be increased if necessary. Figure 2 shows examples of calibration plots where the ozone was varied and the graphite scrubber was

alternately replaced with a conventional packed-bed Hopcalite scrubber. To test for reproducibility, four different sets of graphite tubes were used within the heater. Slopes of heated graphite vs. Hopcalite ozone measurements ranged from 0.96 to 1.01 among the different sets for concentrations up to 500 ppb of ozone. The overall average slope was 0.98. All calibration plots were highly linear with $R^2 > 0.99$ and their slopes were invariant over the graphite scrubber temperature range of 100-130°C.


## 3.2      Long-term reliability of the graphite scrubber

A primary concern with carbon-based scrubbers is that as the surface of the solid-phase carbon oxidizes, it becomes less reactive toward ozone. In principle, the carbon is oxidized to CO and $CO_2$,



which desorb to the gas phase and are swept away in the case of a flowing analyzer. However, the

oxidation to CO and $CO_2$ in the presence of water vapor typically occurs in several steps through various oxygenated intermediates (i.e., epoxides, alcohols, aldehydes, and ketones), and it is likely that at least some of the carbon lost is in the form of small oxygenated organics such as formaldehyde, acetone and acetaldehyde. These do not present a problem as long as they do not absorb appreciably at 253.7 nm. Ellis and Tometz (1972) were among the first to note that upon exposure to high ozone concentrations,

cocoanut charcoal becomes less efficient at removing ozone as it oxidizes. This suggests that oxygenated hydrocarbons on the graphite surface or oxygen functionalities incorporated within the graphite structure have slower ozone reaction rates. A loss in reactivity toward ozone has also been observed for graphite at high levels of ozone (Razumovskii et al., 2007). We also observed this loss in ozone removal efficiency at room temperature and found that heating the graphite scrubber remedies this problem. At elevated

temperatures (100°C), continuous exposure of 150 ppb of ozone for 22 hours in the single-beam Model 202 showed no loss of the graphite's ability to destroy ozone.

However, incorporation of the graphite scrubber into a dual-beam Model 205 led to unexpected complications. We found that even at 100°C, it was necessary to reduce the flow rate to the minimum required to flush the absorbance cells in order to observe 100% ozone removal. Furthermore, overnight

exposure of the graphite tubes to 150 ppb of $O_3$ in the dual-beam Model 205 resulted in a 10% loss of the ozone removal efficiency. Exposure to higher ozone concentrations (300-700 ppb) in the dual-beam Model 205 resulted in faster temporal decays which leveled off at ~ 85% ozone removal efficiency. The graphite's capacity to destroy ozone could be restored by periodically heating the graphite to 150°C for 20-30 minutes or by removal of the ozone exposure while continuing with normal analyzer operation for

3-6 hours. Ellis and Tometz (1972) also had noted that removal of charcoal from ozone exposure for several hours resulted in a regeneration of the surface reactivity toward ozone.

These observations suggest a competition between ozone oxidizing the graphite surface (making less reactive species at the surface) and loss of these oxidized carbon-containing compounds to the gas phase (which presumably exposes new, less oxidized surface area). This competition helps explain why

the single-beam Model 202 is less susceptible to losing reactivity toward ozone since the scrubber is only exposed to ozone flow half of the time (during the $I_o$ measurement). During the $I$-measurement, there is no flow through the scrubber, but oxidized compounds can still desorb and regenerate the surface. In the dual-beam monitor, the graphite is continuously exposed to a flow of ozone as it is always directed to one of the two cells for an $I_o$ measurement. As a result, the surface is being oxidized faster than the graphite

can regenerate. One also expects this competition to be dependent upon the temperature of the scrubber and the flow rate through it.



Therefore, before the graphite scrubber can be used in long-term continuous monitoring, it is necessary to fully understand its limitations in terms of its loss of ozone removal efficiency. To further characterize the graphite scrubber, we undertook extended ambient ozone measurements on the 2B Technologies roof in Boulder, CO during August and September of 2015. A single-beam Model 202 was equipped with our heated graphite scrubber (denoted as 202-G) and sampled from the same Teflon inlet line as an FEM (Federal Equivalent Method) Model 205 dual-beam analyzer with a conventional packed-bed Hopcalite scrubber (205-H). The Model 205 was equipped with Nafion for humidity equilibration; however, Nafion was removed from the Model 202/graphite scrubber analyzer (humidity interference is discussed in Section 3.3). Calibrations were performed on both instruments prior to and periodically throughout the comparison with a Model 306 ozone calibrator. Although the 205-H may be more susceptible to interfering compounds (described in following section), the anticipated bias is < 3 ppb

We initially evaluated the graphite scrubber temperature to 100°C since we had observed no loss of ozone removal efficiency during lab ozone exposure experiments with our single-beam ozone instrument. Ambient measurements were made from August 8-24, 2015. Ozone data was averaged to 5 minutes from both instruments (205-H and 202-G) and compared each day. The daily regression slopes were near unity (0.99 ± 0.02) for the first 12 days; however, on the 13th day, comparisons showed that the 202-G analyzer underestimated peak afternoon ambient ozone by ~ 8% and the 202-G continued to drop over the next three days yielding a regression plot slope of ~ 0.85 versus the 205-H.. This is shown in Figure 3c, where the slopes of the daily comparison between the 202-G and 205-H are shown as a function of the cumulative ozone exposure (in ppm-hr). The calibration of the 202-G was rechecked and confirmed the scrubber only removed 86 ± 2% of the ozone passing through it.

The graphite was regenerated by heating to 150°C for 1 hour with no ozone present and the calibration slope returned to 1.01 ± 0.02. The 202-G was then returned to sampling but the graphite scrubber was maintained at 130°C – the highest temperature without inducing significant VOC emission from the graphite tubes as described in Section 3.1. Post-regeneration, the analyzers were compared again from August 24 to September 30, 2015 (38 days). Figure 3a shows a diurnal plot of the ozone measured by the two analyzers on Sept. 26, 2015, near the end of the intercomparison. It is clear from this plot and the regression plot of the same data in Figure 3b that there was excellent agreement between the two instruments. The daily intercomparison slope vs. ozone exposure in Figure 3c confirms there was no graphite scrubber efficiency loss after 38 days (or 30 ppm-hr ozone exposure). A recheck of the analyzer calibrations found them to be within the uncertainty of the initial calibrations. Therefore, it appears the graphite scrubber must be maintained at temperatures greater than 100°C for reliable long-term continuous monitoring. A temperature of 130°C appears to be optimal for maintaining the balance between oxidation of the graphite by ozone and the loss of oxidized carbon from the surfaces to





regenerate fresh carbon. It should be noted that these results were for a single-beam ozone instrument, where the scrubber is exposed to ozone only during the $I_o$ phase of the measurement cycle. It may be necessary to use a larger surface area or periodic regeneration heating cycles to ensure the reliability of the graphite scrubber in dual-beam analyzers; research is ongoing in this area.


### 3.3 Interference testing

As mentioned earlier, carbon-based scrubbers (typically charcoal or activated carbon) remove ozone from air, but they also effectively remove many other species such as VOCs and water and thus would introduce interferences in ozone measurements. However, the more ordered planar structure of graphite and the greatly reduced surface area provide fewer sites for adsorption of interfering species.
The results of our interference testing described below were performed with the graphite scrubber heated to 100°C. Heating the graphite scrubber to 130°C (the more optimal temperature found in our subsequent longevity testing described in the previous section) would likely reduce adsorption of water, mercury and aromatic VOCs. Hence, these results can be viewed as a conservative estimate of the expected behavior
of the graphite scrubber in reducing interferences in ozone measurements.

#### 3.3.1 Water vapor

Previous studies have shown that rapid humidity changes can cause large apparent ozone absorptions (equivalent to tens of ppb) in most commercial ozone analyzers (Kleindeinst et al., 1993; Wilson and Birks, 2006; Spicer et al., 2010). Wilson and Birks (2006) explained this behavior as being
due to changes in the refractive index at the surface of the optical cell resulting from varying amounts of water vapor adsorbed to cell walls. Initial experiments with a Model 202 analyzer equipped with a conventional Hopcalite scrubber but no Nafion tube, indicated large signal changes coinciding with changes in humidity (Figure 4, black dashed line). A large positive interference was observed as humidity was rapidly increased (uptake of water by the scrubber) and an opposite negative interference
seen upon changing from humid to drier air (release of water from the scrubber). Figure 4 (magenta line) confirms that placement of a Nafion tube at the entrance to the optical cell to equilibrate the humidity with surrounding air (same humidity for $I$ and $I_o$) effectively eliminates the water vapor interference, as shown earlier by Wilson and Birks (2006). A drawback of this approach is that Nafion tubing is expensive, and the length of tubing required to attain adequate equilibration can place limits on its use in
miniaturized analyzers or can introduce flow restrictions, thus making it impractical in some applications.

The graphite scrubber proposed also mitigates artefacts caused by changing water vapor concentrations. As seen in Figure 4, the initial large change in humidity does cause an apparent absorption of ~ 10 ppb of "O$_3$" with the graphite scrubber in-line. It appears that some water is initially taken up by the graphite, likely due to the somewhat porous nature of the graphite structure. However,



water content within the graphite scrubber equilibrates within about 2 minutes and returns to its original

value, whereas the Hopcalite trap remains compromised for > 15 minutes. Subsequent removal of the

high water vapor concentration leads to a negative response as water vapor desorbs from each scrubber.

But again, the recovery of the graphite scrubber is relatively rapid compared to the Hopcalite, so although

there is some water uptake, equilibration is rapid enough to exceed typical humidity changes and 30-60

minute ozone averages will be relatively unaffected.

Variation of relative humidity did indicate a slight water dependency in the graphite scrubber and

exhibited some differences between different graphite tubes. Figure 5 illustrates this behavior, showing

two different sets of graphite tubes (4 tubes in parallel, 3-in long). Set #1 indicated about a +2.5 ppb

change from dry to near-saturated air; whereas Set #2 was essentially independent of the humidity.

Given graphite's hydrophobic nature, these differences are perplexing, but could be due to differing

degrees of oxidation on the graphite surface or on binders within the graphite, making the surface slightly

more hydrophilic. The graphite scrubber's low sensitivity to humidity was also confirmed in our ambient

measurements (Section 3.2) where we observed no correlation between ambient humidity changes and

concentration differences between the graphite equipped Model 202-G and the Hopcalite and Nafion

equipped Model 205-H.

### 3.3.2 Mercury

Gaseous elemental mercury concentrations are typically less than a few ppt in ambient air, but

mercury poses a problem in ozone photometers because typically the Hg atomic absorption line is used as

the photometer's excitation source (Hg line at 253.7 nm). Birks et al. (2009) estimate that common ozone

photometers could be 1860 times more sensitive to elemental mercury than ozone (Selectivity = 1.86 ppb

$O_3$/ppt Hg) depending on how efficiently the scrubber removes mercury. Recent measurements in our

laboratory showed that mercury absorbs ~1350 times more strongly than ozone in the Model 202 Ozone

Monitor using the packed-bed Hopcalite scrubber (Birks, 2016 unpublished data). There is also evidence

suggesting that the efficiency of mercury uptake in several conventional scrubbers may depend on

humidity (U.S-EPA, 1999; Spicer et al., 2010), implying that these may act as a temporary reservoir

capable of releasing mercury at later times following humidity changes.

In order to examine possible interferences due to elemental mercury, varying concentrations of

mercury vapor were added to the Model 202 ozone monitor and responses were monitored (in apparent

ppb of ozone) while the various graphite and conventional scrubbers were interchanged. As we did not

have an independent measurement of mercury vapor concentrations, all measurements were made relative

to the conventional packed-bed Hopcalite scrubber used in the 2B Technologies analyzers. For different

levels of mercury additions, Figure 6 shows the apparent ozone measured by the photometer as various

scrubbers were alternated with the packed-bed Hopcalite. Five different sets of graphite tubes (set = 4





tubes in parallel, each 3-in long) were tested and the slopes of plots of apparent ozone concentration vs.

those obtained for the Hopcalite scrubber ranged from < 0.01 to 0.03 (median set is shown in Figure 6).

One set showed no measurable response to mercury up to an apparent ozone concentration of 2,200 ppb

for the Hopcalite scrubber.  Noting above our recent observation that mercury absorbs 1350-fold stronger

than ozone (or 1.35 ppb $O_3$/ppt Hg), this leads to a response ratio of $\leq 0.04$ ppb $O_3$/ppt Hg for the graphite

scrubber. Measurements made at 22% and 80% RH exhibited no differences (Figure 6).  In addition,

measurements made with and without Nafion tubing inserted before the optical cell gave identical results,

suggesting that mercury is neither lost nor interacts significantly with Nafion.

The heated silver wool scrubber exhibited the largest mercury response, with a slope slightly

greater than unity compared to Hopcalite (Fig. 6).  Silver is known to rapidly form an amalgam with

mercury (Dumarey et al., 1985) which likely explains this large response.  Previous studies have shown

that both the 2B Technologies Hopcalite scrubber and heated silver wool scrubber remove mercury vapor

efficiently – reporting response ratios between 0.6 to 1.1 ppb $O_3$/ppt Hg (Spicer et al., 2010; U.S.-EPA,

1999).  We report a response ratio of 1.49 ppb $O_3$/ppt Hg for the heated silver wool assuming 1.35 ppb

$O_3$/ppt Hg for the Hopcalite scrubber – approximately 1/3 higher than the previous studies.  The

sensitivity to Hg depends on the width of the emission line of the lamp (Birks et al., 2009), which may

vary from instrument to instrument.

Both the Teledyne-API and Thermo Electron scrubbers use screens coated with $MnO_2$ and

showed less interference from mercury vapor compared to the packed-bed Hopcalite scrubber (slopes of

0.18, and 0.065).  This yields response ratios of 0.24 and 0.09 ppb $O_3$/ppt Hg for the Teledyne-API and

Thermo Electron scrubbers, respectively.   Smaller responses are likely due to the large reduction in

scrubber surface area by use of the coated screens.  Measurements made at both high ($\geq 80\%$) and low

(20%) relative humidity showed no significant differences, although the considerable scatter in the data

likely obscures any small humidity dependencies.  The interference of mercury in ozone photometers with

$MnO_2$-based scrubbers has been the subject of several previous investigations.  The U.S.-EPA (1999)

reported response ratios in the range 0.12 to 0.24 ppb $O_3$/ppt Hg using a Thermo Electron scrubber.  This

study also suggested a lowering of the response ratio with increasing humidity.  Higher response ratios

(0.87 and 0.6 ppb $O_3$/ppt Hg) have also been reported in dry air (Li et al., 2006; Spicer et al., 2010).

Spicer et al. (2010) also reported that the response ratio decreased by a factor of 2 when using humidified

air (RH ~ 80%), which is closer to both our results and those of the EPA.  As will be shown for VOCs in

the next section, it is likely that even a small amount of water vapor can make a significant change in the

adsorption properties of these solid-phase scrubbers; thus the higher response ratios determined in dry air

may not be applicable to sampling of ambient air.  The use of a heated graphite scrubber can reduce the



interference in ozone photometers from mercury vapor by a factor of ≥ 30 over the packed-bed Hopcalite scrubber and by a factor of ≥ 2 over the best solid-phase scrubber.

This low response ratio is only matched by using the gas-phase nitric oxide (NO) titration as a means of scrubbing ozone. In the recently introduced 2B Technologies Model 211 Scrubberless Ozone Monitor™, gas-phase NO titration eliminates the need for a solid-phase scrubber, and effectively eliminates interferences from both UV-absorbing species and water vapor (Birks et al., 2013). However, a source gas of NO is required, reducing the portability of the instrument and adding long-term cost. The heated graphite scrubber avoids these disadvantages.

In general, mercury is not expected to be a significant interference in outdoor ambient air. Although there are some elevated measurements of mercury vapor in urban areas of Asia (up to ~ 3 ppt: Fang et al., 2004; Liu et al., 2002; Kim and Kim, 2002), concentrations typically range from 0.2 (background) to 0.5 (urban) ppt (Obrist et al., 2008; Weiss-Penzias et al., 2003; Denis et al., 2006). However gas-phase mercury can reach higher levels near industrial or mining sources. Elevated mercury

levels have also been shown to be present for indoor air (Carpi and Chen, 2001; Garetano et al., 2006). Because indoor ozone concentrations are typically low (~ 20-80% of outdoor air, U.S. EPA, 2016), mercury poses a larger potential problem for indoor air quality monitoring of ozone. For example, Carpi and Chen (2001) report indoor mercury concentrations of up to 520 ng m$^{-3}$ (~ 63 ppt). This would correspond to measured absorbances equivalent to ~ 85 ppb $O_3$ with the packed-bed Hopcalite, ~ 6-15

ppb $O_3$ with conventional $MnO_2$ screens, and < 2.6 ppb $O_3$ using the heated graphite.

### 3.3.3 Volatile Organic Compounds (VOCs)

     Interferences from volatile organic compounds are more complicated due to the wide variety of compounds and their properties. There are many diverse atmospheric VOCs having a wide range of 253.7 nm absorption cross sections, volatilities, and potential responses to humidity. Along with the

VOC absorption cross sections, Table 1 also lists boiling points and room temperature vapor pressures as indicators of volatility. Aromatic VOCs are the most important when discussing interferences in ozone monitors since the conjugation between the carbon-carbon double bonds both shifts the absorption spectrum to longer wavelengths and results in larger UV cross sections. Addition of various functional groups (hydroxyl, aldehyde, nitro, etc.) to the aromatic ring can further increase the absorption cross

section. Furthermore, substitutions often result in lower volatility, making these species more likely to be retained in a solid-phase scrubber.

     Interferences from VOCs typically result from two different mechanisms: (1) adsorption of the compound by the scrubber, resulting in a positive absorbance measurement ($I_o > I$), and (2) subsequent later desorption of that compound from the scrubber due to eventual elution and/or changes in

temperature or humidity that yield a negative absorption ($I_o < I$). These two processes are complex





functions of environmental conditions (temperature and humidity), volatility and chemical properties of the specific VOC, as well as the reactivity and surface area of the scrubber material.

Past studies of VOC interferences have typically been of three types: (1) measurement of total net ozone biases during irradiation of simulated atmospheres in smog chambers (Kleindeist et al., 1993; Leston et al., 2005); (2) simultaneous atmospheric measurements by different analyzers using different measurement techniques (Ollison et al., 2013; Dunlea et al., 2006); and (3) addition of selected VOCs to ozone analyzers to determine the interference from that particular VOC (Grosjean and Harrison, 1985; Spicer et al., 2010). All of these approaches contribute to our understanding of the significance of VOC interferences in ozone monitors; however, we have chosen the latter approach to initially evaluate our heated graphite scrubber because it is a more direct approach and reproducible signals can be easily quantified. We chose p-xylene, phenol, p-tolualdehyde, and o-nitrophenol, as test VOCs since they have reported 253.7-nm absorption cross sections ranging from $5 \times 10^{-19}$ (p-xylene) to $2.15 \times 10^{-17}$ (o-nitrophenol) cm$^2$ molecule$^{-1}$ (Keller-Rudek et al., 2013) and boiling points from 138 to 279°C.

*VOC Interferences in Dry Air.* Initial experiments examined VOC interferences in dry air (RH < 3%) in the absence of ozone using a single-beam Model 202 monitor equipped with the various scrubbers described above. No Nafion tubing was installed upstream of the optical cell in the monitor. VOC concentrations were measured by sampling the same air stream into the inlet of a GC-FID. Upon VOC addition, apparent ozone absorbances were observed to reach steady values within the initial 2-4 minutes with both the Hopcalite and MnO$_2$-based scrubbers. With the heated graphite scrubber, there was a rapid increase in apparent ozone upon addition, followed by a decay to steady values after 3 minutes. Average apparent ozone concentrations were obtained after 5 minutes of exposure and plotted vs. VOC concentrations measured with the GC-FID (Figure 7). There were no signs of surface saturation effects, as indicated by the linearity of the apparent ozone with VOC concentration plots. For all the compounds tested, the heated graphite scrubbers showed less apparent ozone. The slopes of the plots in Figure 7 are given in Table 2. In all cases, the slopes were a factor of 2.5 to 20 less for the graphite scrubber than for Hopcalite or the MnO$_2$ scrubbers, indicating greatly reduced VOC uptake by the graphite. Data from two sets of graphite tubes (4 tubes in parallel) gave identical results for both p-xylene and phenol; however, both p-tolualdehyde and o-nitrophenol did exhibit slightly different apparent ozone values from these two different sets of graphite tubes. Responses for three other sets of graphite tubes were also tested against the Hopcalite scrubber and gave absorbances within the range of those shown in the figure. It remains unclear why different tube sets show slightly different behavior, but, again, it likely lies in the degree of graphite surface oxidation.



In dry air both Hopcalite and $MnO_2$ scrubbers exhibited similar responses for all the compounds tested. If the scrubbers remove 100% of the VOC compound, then the slope of these plots should be the
inverse of the selectivity ratio reported in Table 1:

$$\frac{1}{S} = Slope = \frac{\sigma_{VOC-254nm}}{\sigma_{O3-254nm}} \text{ (in ppb O}_3\text{/ppb VOC)} \tag{2}$$

There is good agreement in the theoretical slope and that observed for p-xylene (see Table 2), suggesting both Hopcalite and $MnO_2$-based scrubbers effectively scavenge all of the p-xylene in dry air. However, the slopes for phenol, p-tolualdehyde, and o-nitrophenol were all significantly higher by factors of 1.4,
3.3, and 4.1, respectively, than those computed from previously reported absorption cross sections. As this would require >100% removal by the scrubber, this can only be explained either by errors in the previously reported VOC cross sections or an underestimation in the VOC concentrations via the GC technique. Since the error tends to increase with volatility, the latter is more likely as it becomes more difficult to quantitatively desorb the VOC from the Tenax pre-concentration trap used in the GC.
Furthermore, if our o-nitrophenol concentrations were correct, this suggests a 253.7-nm absorption cross section of nearly $9 \times 10^{-17}$ cm$^2$ molecule$^{-1}$, which seems inexplicably large.

We did not pursue a more accurate GC-FID VOC quantitation as the heated graphite scrubber can still be assessed on a relative basis to the conventional scrubbers from the ratios of the slopes in Figure 7. Assuming that the packed bed Hopcalite scrubber exhibits the theoretical selectivity factor based on
absorption cross sections for each VOC tested (reported in Table 1), selectivity ratios of 83, 28, ≥ 12, and ≥ 1.5 ppb VOC/ppb $O_3$ were measured for p-xylene, phenol, p-tolualdehyde, and o-nitrophenol, respectively, for the heated graphite scrubber. In other words, it would require a p-xylene concentration of 83 ppb to produce a signal of 1 ppb $O_3$ in an analyzer using a heated graphite scrubber compared to a concentration of 20 ppb for the Hopcalite or $MnO_2$ scrubbers.
We observed less than 10% difference between the response ratios obtained with the Hopcalite and the two $MnO_2$ thin-film screen scrubbers for both p-tolualdehyde and o-nitrophenol in dry air. Spicer et al. (2010) report values of 0.8 and 0.0 to 0.5 ppb $O_3$/ppb p-tolualdehyde in dry air when using the $MnO_2$ and Hopcalite scrubbers, respectively, a difference of ≥ 30%. They also report values of 2.2 and 1.1 to 1.8 ppb $O_3$/ppb o-nitrophenol for the $MnO_2$ and Hopcalite scrubbers. This leads to differences
ranging from 20% to a factor of two, contrary to that observed in the current study. However, concentration levels of interfering VOCs were quite low in the Spicer et al. (2010) study, ranging from 7.6 to 14 ppb and their measured apparent ozone mixing ratios were < 15 ppb. At these levels, small signal drifts, or even the typical precision of ± 1 to 2 ppb in the ozone analyzers impart significant measurement uncertainty. It should be noted that the value of 0.8 ppb $O_3$/ppb p-tolualdehyde reported in





Spicer et al. (2010) for the MnO$_2$ scrubber is also above the maximum theoretical response based on the reported absorption cross sections by a factor of 1.7 (1/S = 0.48, Table 1).

Zdanevitch (2002) reported a higher response for phenol and other VOCs studied with ozone present and suggested that photochemical production of stronger-absorbing compounds within the optical cell may be responsible. Our experiments with p-tolualdehyde and p-xylene failed to show such evidence

for photochemical behavior. Absorbances recorded with both ozone and VOC present were equivalent to the sum of the absorbances when these components were measured individually within the precision of the analyzer. This supports calculations that suggest negligible photochemistry can occur due to the low photon flux and short residence time in the 2B Technologies Model 202.

*VOC interferences in humid air.* Even at low humidity levels, water vapor is adsorbed to some

extent by most of the conventional solid-phase scrubbers. This changes the surface characteristics of the solid phase which, in turn, can alter the adsorption and desorption of other compounds. This is also dependent upon the properties of the species of interest – especially the particular compound's volatility. For the most volatile compound studied, p-xylene, even low levels of humidity caused the observed interference (apparent ozone) to decrease over time. Figure 8a shows the temporal behavior of the

apparent ozone absorption upon addition of ~ 1 ppm of p-xylene in moderately humid ozone-free air with an analyzer equipped with a conventional packed-bed Hopcalite scrubber and Nafion. There is an initial spike in the signal that then decays over time. The inset of Figure 8a shows how this decay varied as a function of RH. In each case, p-xylene was added to generate an apparent ozone signal of ~ 80 ppb in dry air. The humidity was then changed within 20 seconds and the absorbance decay was recorded. The

apparent ozone decay was faster at higher humidity, suggesting that the presence of water causes less p-xylene to be retained in the scrubber.

However, even at these higher water vapor concentrations the scrubber still accumulated p-xylene. This is apparent upon turning off the p-xylene source, as a large negative absorbance was then observed (Figure 8a). p-Xylene was now slowly desorbing from the scrubber material, thus absorbing

more light during the $I_o$ measurement relative to $I$. This slow emission was also humidity dependent – becoming faster at higher relative humidity (data not shown). Scrubbers from the Thermo Electron and Teledyne-API instruments show similar behavior except that the time response is much faster (< 2 min., Figure 8a), presumably due to the lower surface area of scrubber material. In fact, these scrubbers were observed to pass p-xylene nearly quantitatively, similar to the graphite scrubber after about 5 minutes as

long as RH > 10%. These scrubbers also showed rapid release of the p-xylene once the VOC exposure was discontinued. However, the equilibration is rapid enough that, even at low humidity, the effect of xylenes on hourly averaged ambient ozone measurements is likely negligible with these low surface area scrubbers.



VOC responses with the heated graphite scrubber tend to be relatively insensitive to changes in
relative humidity. Figure 9 shows a 1 ppm p-xylene addition to an ozone analyzer initially equipped with
a Hopcalite scrubber at low RH (9 %), giving an apparent $O_3$ concentration of 67 ppb. The Hopcalite was
then exchanged for a heated graphite scrubber and the relative humidity varied over the following 95
minutes where there is little to no RH-driven change in the observed absorbance. A plot of the averaged
signal versus relative humidity yields a slope of 0.018 ppb apparent $O_3$/% RH, where a humidity change
from 20 to 80% only alters the apparent ozone signal from 1 ppm of p-xylene by +1 ppb.

As the VOC becomes less volatile, the advantages of the heated graphite scrubber become more
significant. Experiments with p-tolualdehyde indicated that a small amount of humidity (RH = 22%) did
not alter the behavior from that observed in dry air for any of the various scrubbers. However, as RH
increased to 75%, the coated $MnO_2$ screen scrubbers showed a large increase, followed by a slow decay
over 20-30 minutes (Figure 8b) – behavior akin to the packed-bed Hopcalite/p-xylene exposure discussed
previously. These scrubbers also exhibited a larger negative absorbance when the p-tolualdehyde flow
was removed indicating release of adsorbed p-tolualdehyde post-exposure, similar to the Hopcalite/p-
xylene temporal profiles. For phenol, the selectivity for both the Hopcalite and $MnO_2$ scrubbers remained
constant regardless of humidity (data not shown). Even at RH = 75%, the Hopcalite and $MnO_2$ scrubbers
behaved similar to dry air, quantitatively removing phenol and showing no decay in the apparent $O_3$
absorbance over at least 10 minutes. We did not examine the behavior of o-nitrophenol with humidity,
but presume it would be similar to that of phenol.

Although the conventional $MnO_2$ scrubbers with low surface area appear to be compatible with
the more volatile aromatics like xylene, their propensity to have uptake and re-emission of less volatile
compounds dependent upon humidity levels is problematic in estimating ozone monitor biases. The
selectivity ratio described above is now a function of humidity which varies in time. The wide variety of
different atmospheric VOCs and their varying properties suggest that it will be nearly impossible to
quantitatively model or estimate whether these scrubbers are consuming or releasing compounds, i.e.,
whether ozone monitors are biased negatively or positively. For example, Spicer et al. (2010) reported a
decrease in the percent response ratio for the $MnO_2$ scrubber from 0.8 to 0.1 ppb p-tolualdehyde/ppb $O_3$
going from dry to humid air. This lowering of the p-tolualdehyde absorbance with RH was also observed
in this study, but was dependent upon exposure time such that a single selectivity ratio cannot be used. In
contrast to the conventional solid-phase scrubbers, VOC interferences observed with the heated graphite
scrubber are virtually insensitive to humidity changes for all the compounds studied. Although the
graphite scrubber does retain a small fraction of the VOCs, the lack of a humidity dependence makes
potential ozone monitor biases easier to predict.



## 4   Conclusions

A new solid-phase scrubber based on graphite for use in conventional ozone photometers was investigated and found to effectively remove ozone, while offering significant advantages over current ozone scrubbers with regard to species that typically interfere in ozone measurements. When heated to 100-130°C, the graphite scrubber quantitatively removed ozone up to 500 ppb – well above the concentration levels required for most ambient or indoor monitoring studies. After two weeks of ambient measurements in a single-beam instrument slow oxidation of the graphite surface reduced its ability to completely remove ozone when operated at the lower end of the temperature range studied ($\leq 100°C$). However, when operated at 130°C, no loss of ozone removal efficiency was observed up to exposures of 30 ppm-hr over 38 days, and agreement with a conventional FEM analyzer was excellent throughout the time period. The higher temperature likely facilitates removal of lower reactivity oxidized species from the graphite surface either by enhancing desorption or the reaction rate with ozone. Therefore, long-term measurements require continuous operation near the maximum of 130°C or periodic regeneration of the scrubber by heating it to 150°C for 20-30 minutes. Caution must also be exercised using the graphite scrubber in dual-beam ozone analyzers where the scrubber is continuously exposed to ozone. This led to a relatively faster loss of the ozone removal efficiency (10% after ~ 3 ppm-hr $O_3$ exposures), and thus the current scrubber design is not recommended for dual beam analyzers.

The major advantage of the heated graphite scrubber is its ability to pass many of the known atmospheric interferences that plague conventional solid-phase ozone scrubbers. Elimination of small biases of a few ppb of ozone can be critical in marginal compliance situations where ambient ozone is near the NAAQS. Water effects on ozone measurements (through refractive index changes in the optical cell) were virtually eliminated by use of the graphite scrubber, thereby eliminating the need for humidity equilibration using Nafion. Gas-phase mercury interference was $\leq 3\%$ of a packed-bed Hopcalite scrubber, and at least a factor of two better than other conventional $MnO_2$ coated-screen scrubbers. The graphite scrubber was also observed to consistently exhibit smaller interferences in dry air for a series of substituted aromatic compounds ranging in volatility and extinction coefficient. The apparent ozone absorptions measured with the graphite scrubber were typically a factor of 2.5 to 20 smaller than those with the conventional Hopcalite or $MnO_2$ scrubbers. Our results suggest that to generate an absorbance equivalent to 1 ppb of ozone would necessitate concentrations of 83 ppb of p-xylene, 28 ppb of phenol, $\geq$ 12 ppb of p-tolualdehyde, $\geq$ 1.5 ppb of o-nitrophenol, or $\geq$ 25 ppt of mercury vapor.

The small VOC interference observed with the graphite scrubber was also insensitive to humidity level. Although the conventional commercial solid-phase scrubbers tended to remove all of the aromatic VOCs studied in dry air, these scrubbers tended to show less interferences at higher humidity levels (i.e., less aromatic compounds retained in the scrubber). This lowering of the interference was tied to the





particular VOC's intrinsic volatility and the surface area of the solid phase scrubber. Low volatility compounds (such as phenol) were retained regardless of humidity level, whereas compounds with higher volatility (p-xylene) were nearly quantitatively passed by the scrubbers using coated screens with low surface area. For compounds that were intermediate in volatility, such as p-tolualdehyde, the conventional commercial scrubbers quantitatively removed the VOC at low humidity (< 40% RH), but the coated-screen scrubbers showed a slow equilibration at high water vapor concentrations (> 70% RH, $t_{equil}$ ~ 30 min.). These also released p-tolualdehyde upon ending exposure to the compound. These results strongly suggest that these scrubbers tend to accumulate VOC under drier conditions (creating a positive bias in ozone analyzers) and then subsequently re-release them as humidity is increased (creating the possibility of a negative bias). This complex behavior involving humidity and volatility is avoided by use of the heated graphite scrubber.

Although more work is needed to determine whether the graphite scrubber can replace conventional solid-phase scrubbers for long-term compliance monitoring, both the lab and field studies presented here suggest that the heated graphite scrubber has the potential to provide more accurate ozone concentration measurements by reducing conventional photometer susceptibility to many common interferences. Reducing interferences in ozone measurements is especially important for ozone monitoring in areas of marginal non-compliance and in exposure monitoring in indoor settings.

## 5 Acknowledgements

The authors would like to acknowledge Dr. John Ortega at the National Center for Atmospheric Research for assistance with GC/FID measurements. We would also like to thank Dr. Will Ollison for his advice and critique of this manuscript.

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





**Table 1.** Properties of VOCs and other atmospheric species that are potential interferents in UV absorption-based ozone measurements.

| Interferent (INT) | $\sigma_{254}$,[a] $10^{-17}$ cm$^2$ molecule$^{-1}$ | $b_p$,[b] °C | $e_v$,[b] (25°C) mm Hg | S,[c] ppb INT/ppb O$_3$ |
|---|---|---|---|---|
| **Used in this Study** | | | | |
| p-Xylene | 0.057 | 138 | 8.84 | 20.2 |
| Phenol | 0.16 | 182 | 0.350 | 7.1 |
| p-Tolualdehyde | 0.55 | 205 | 0.250 | 2.09 |
| o-Nitrophenol | 2.15 | 279 | 0.113 | 0.53 |
| Mercury | 1860[d] | 356 | 0.002 | 0.0005 |
| **Typical Atmospheric Constituents** | | | | |
| NO$_2$ | 0.001 | 21 | 741 | 1000 |
| SO$_2$ | 0.014 | -10 | 3000 | 77 |
| Methanol | < 0.00001 | 65 | 127 | > 10000 |
| Acetone | 0.0030 | 56 | 231 | 290 |
| Formaldehyde | 0.00029 | -19 | 3890 | 3960 |
| Acetaldehyde | 0.0015 | 20 | 902 | 590 |
| 1,3-Butadiene | < 0.02 | -4 | 1840 | > 58 |
| Trichloroethene (TCE) | 0.0041 | 87 | 69 | 265 |
| 2-Butanone | 0.0031 | 80 | 90.6 | 370 |
| Acetonitrile | 0.0023 | 82 | 89 | 500 |
| Isoprene | 0.0052 | 34 | 550 | 220 |
| Tetrahydrofuran | < 0.0002 | 66 | 162 | > 5700 |
| Octane | < 0.0002 | 125 | 14.1 | > 5700 |
| Methacrolein | 0.00018 | 69 | 155 | 6400 |
| Methyl Vinyl Ketone | 0.00024 | 81 | 152 | 4790 |
| Acetic Acid | < 0.0006 | 118 | 15.7 | > 1900 |
| α-Pinene | 0.00054 | 155 | 4.75 | 2120 |
| d-Limonene | 0.015 | 177 | 1.55 | 77 |
| **Aromatics** | | | | |
| Benzene | 0.03 | 80 | 94.8 | 38 |
| Toluene | 0.039 | 111 | 28.4 | 29 |
| m-Xylene | 0.042 | 139 | 8.29 | 27 |
| Ethyl Benzene | 0.052 | 136 | 9.6 | 22 |
| Styrene (vinyl benzene) | 1.3 | 145 | 6.4 | 0.88 |
| 1,3,5-Trimethylbenzene | 0.037 | 165 | 2.48 | 31 |
| 1,2,4-Trichlorobenzene | 0.025 | 169 | 2.1 | 46 |
| m-Cresol | 0.108 | 202 | 0.11 | 10.6 |
| Benzaldehyde | 0.09 | 179 | 0.127 | 12.8 |
| Acetophenone | 0.3 | 202 | 0.397 | 3.8 |
| Nitrobenzene | 1.3 | 211 | 0.245 | 0.88 |

a. Absorbance cross sections at 253.7 nm taken from the Keller-Rudek et al., 2013 and related web data base (*http://www.uv-vis-spectral-atlas-mainz.org/*)

b. Boiling points and vapor pressures taken from *http://pubchem.ncbi.nlm.nih.gov/*.

c. Selectivity ratio, $S = \sigma_{O_3} / \sigma_{int}$ at 253.7 nm.

d. From Birks et al. (2009).



**Table 2.** Summary of results from tests of VOC interferences in dry air.

| Volatile Organic Compound | Scrubber | Slope, ppb $O_3$/ppb VOC | $S^a$, ppb VOC / ppb $O_3$ | S, Previous Results |
|---|---|---|---|---|
| **p-xylene** | 2B Hopcalite | 0.056 ± 0.004 | 17.9 | |
| | Thermo Electron (MnO$_2$) | 0.051 ± 0.003 | 19.6 | |
| | Teledyne-API (MnO$_2$.) | 0.046 ± 0.005 | 21.7 | |
| | Heated Graphite | 0.012 ± 0.001 | 83.3 | |
| **phenol** | 2B Hopcalite | 0.21 ± 0.01 | 4.76 | |
| | Thermo Electron (MnO$_2$) | 0.18 ± 0.01 | 5.56 | 7.7-14.3[b] |
| | Teledyne-API (MnO$_2$) | 0.20 ± 0.01 | 5.00 | |
| | Heated Graphite | 0.051 ± 0.004 | 19.6 | |
| **p-tolualdehyde** | 2B Hopcalite | 1.71 ± 0.25 | 0.58 | 0.0-2.0[c] |
| | Thermo Electron (MnO$_2$) | 1.57 ± 0.43 | 0.63 | 1.25[c] |
| | Teledyne-API (MnO$_2$) | 1.54 ± 0.10 | 0.65 | |
| | Heated Graphite | 0.08 ± 0.02 | 12.5 | |
| | | 0.28 ± 0.06[d] | 3.6 | |
| **o-nitrophenol** | 2B Hopcalite | 7.7 ± 0.5 | 0.13 | 0.55-0.91[c] |
| | Thermo Electron (MnO$_2$) | 8.0 ± 0.3 | 0.13 | 0.45[c] |
| | Teledyne-API (MnO$_2$) | 7.4 ± 0.3 | 0.14 | |
| | Heated Graphite | 1.1 ± 0.3 | 0.91 | |
| | | 2.7 ± 0.2[d] | 0.37 | |

a.  S = 1/Slope, measured Selectivity Ratio.
b.  Zdanevitch, 2002.
c.  Spicer et al., 2010.
d.  Results from 2 different sets of graphite tubes.






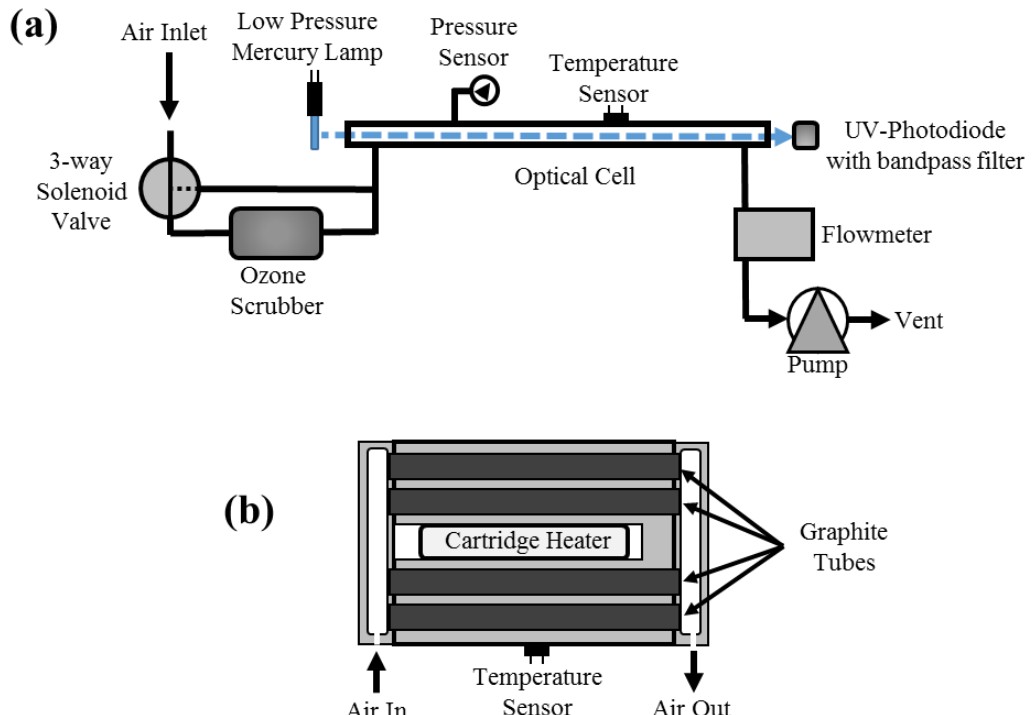

**Figure 1.** (a) Schematic diagram of a typical single-beam UV absorbance monitor for ozone. (b) Schematic of the heated graphite scrubber consisting of four graphite tubes contained within a
temperature-controlled aluminum block.





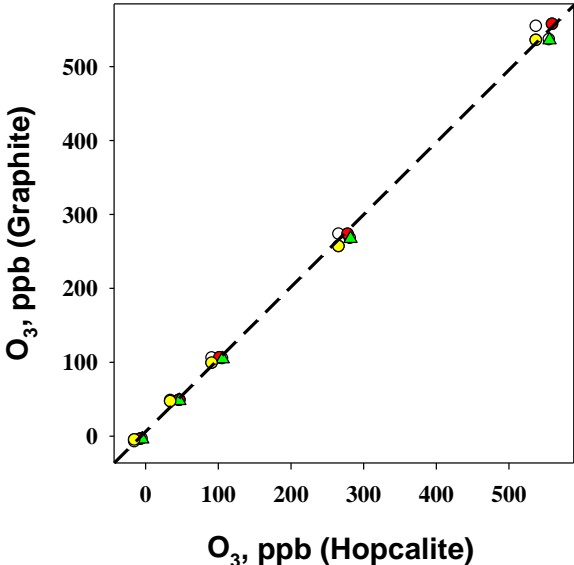

**Figure 2.** Plot of ozone measured with the graphite scrubber (at 100°C) vs. that measured with the conventional Hopcalite scrubber. The different symbols/colors represent different sets of graphite tubes used within the heater during our reproducibility testing. Results were obtained using a 2B Technologies Model 202 single-beam ozone analyzer. The dashed line drawn is the 1:1 line.


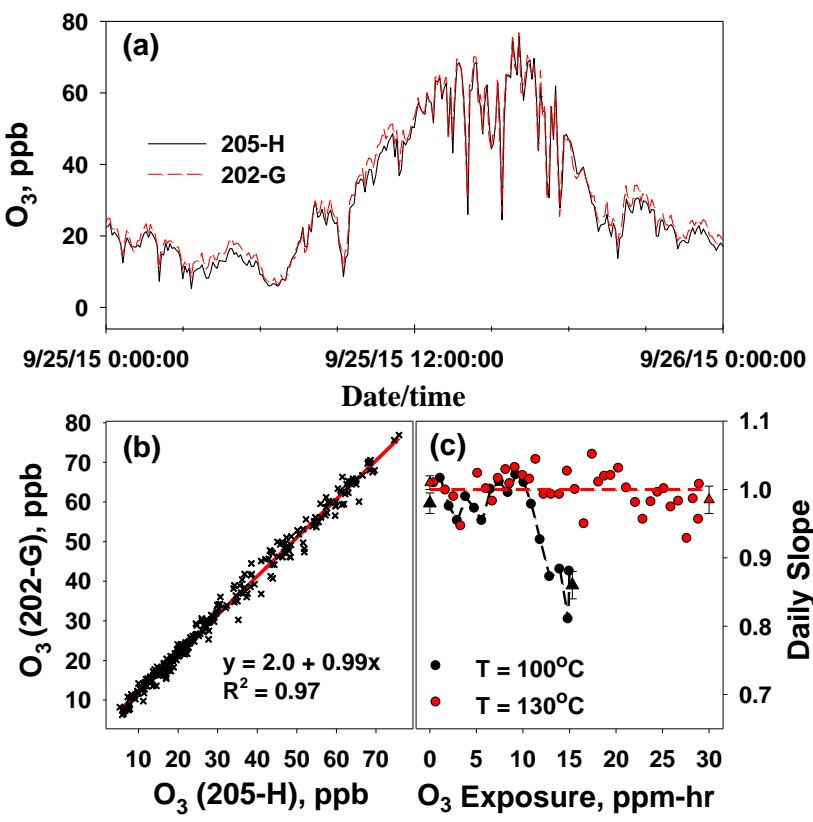

**Figure 3.** (a) Time series of 5-min averaged ozone concentrations measured in Boulder, CO on
September 26, 2015. (b) Correlation plot between the Model 202 (202-G, equipped with graphite
scrubber) and the FEM Model 205 (205-H, Hopcalite scrubber and Nafion) for September 26, 2015. (c)
Plot of the daily correlation plot slope vs. cumulative ozone exposure for the 202-G analyzer. Each point
represents a single day. The graphite scrubber was initially maintained at 100°C, regenerated, and then
operated at 130°C (see text for further explanation). Triangles represent calibration slopes pre- and post-
intercomparison.







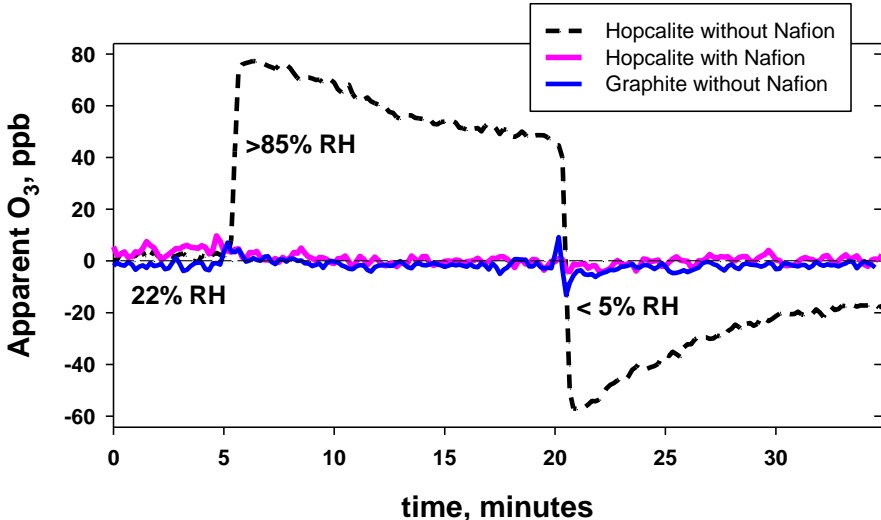

**Figure 4.** Time series showing the response of a Model 202 ozone analyzer to changes in relative humidity with no ozone present and different scrubbers.

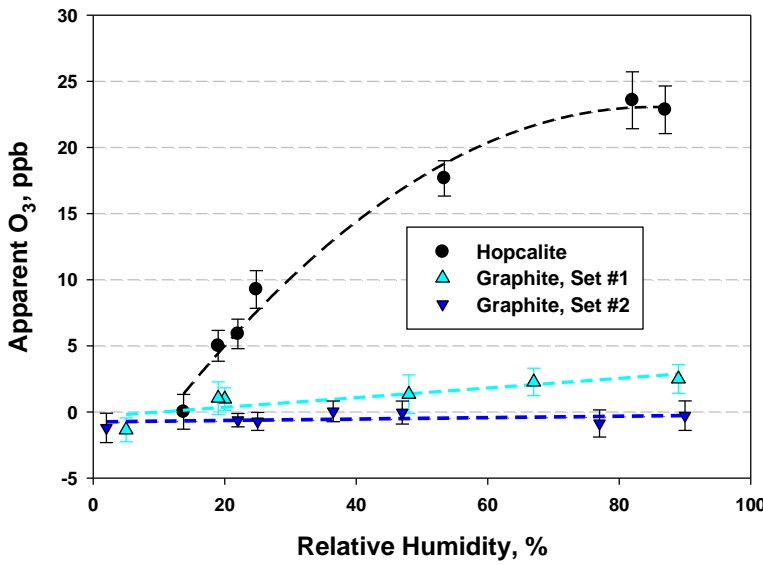

**Figure 5.** Humidity dependence of two different heated (100°C) graphite scrubbers as well as that of a Hopcalite scrubber. The ozone analyzer was 2B Technologies Model 202 single-beam instrument. Data points were 2-minute averages obtained 15 minutes after a change in relative humidity. Error bars are 1σ of the 2-minute average. The experiment was conducted at 20°C. Note that the step changes in humidity were not as large as those shown in Figure 4, thus leading to smaller observed artefact signals for the
Hopcalite scrubber.



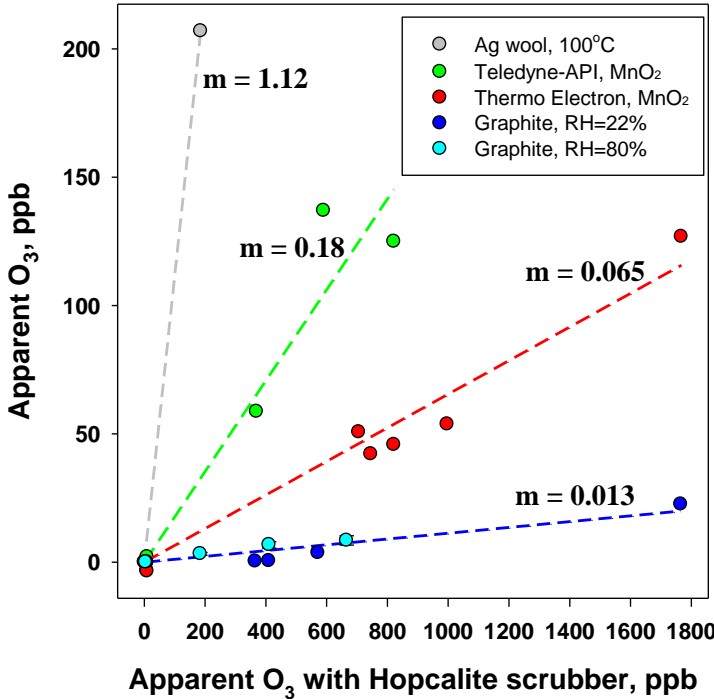

**Figure 6.** Apparent ozone measured in the Model 202 single-beam ozone analyzer using different ozone scrubbers, relative to the standard Hopcalite scrubber, upon addition of mercury vapor to ozone-free air.
Slopes are given by the m-values. The graphite scrubbers were heated to 100°C for these tests.





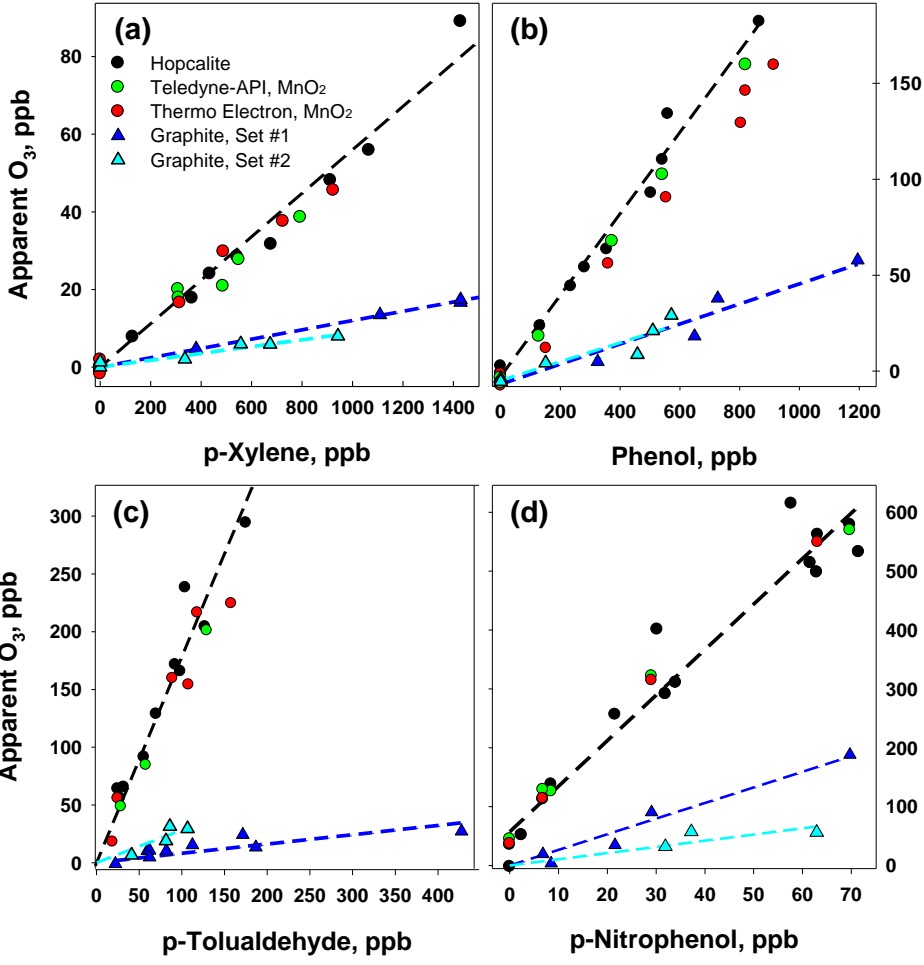


**Figure 7.** Apparent ozone vs. VOC concentration in dry, ozone-free air for different scrubber types and four different VOCs of varying absorption cross section at 253.7 nm. Data from two different sets of graphite tubes are shown (Set #1 and Set #2). Slopes are given in Table 2.



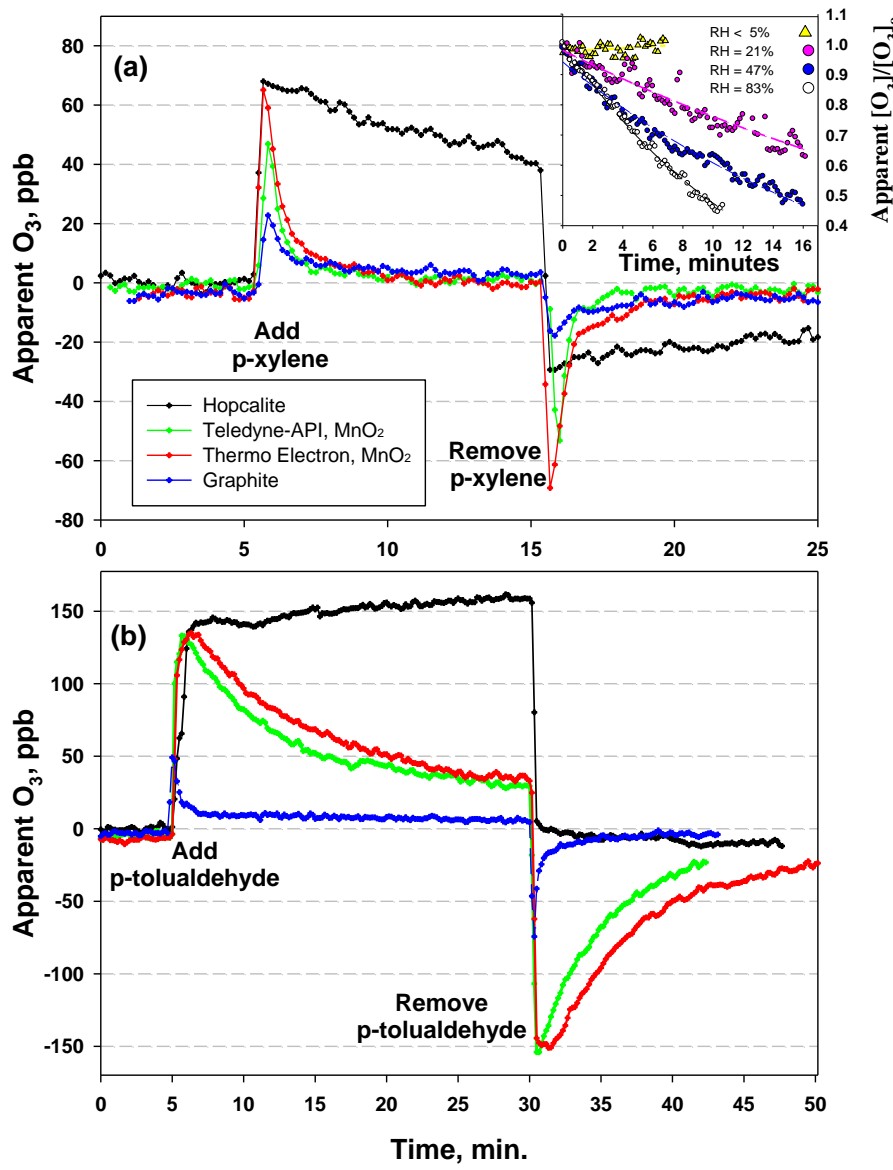


**Figure 8.** (a) Plot of the temporal behavior of the apparent ozone at 38% RH as 1.2 ppm of p-xylene is added (t = 5 min) and then removed (t = 15 min) from ozone-free air for various scrubbers. Inset: Temporal decay of the fractional apparent ozone ($[O_3]_o$ = initial concentration) as a function of relative humidity upon addition of ~ 1.3 ppm of p-xylene to a Model 202 analyzer equipped with a packed-bed
Hopcalite scrubber. (b) Temporal behavior of the apparent ozone at 75% RH as ~ 90 ppb of p-tolualdehyde was added (t = 5 min) and then removed (t = 30 min) for various scrubbers.





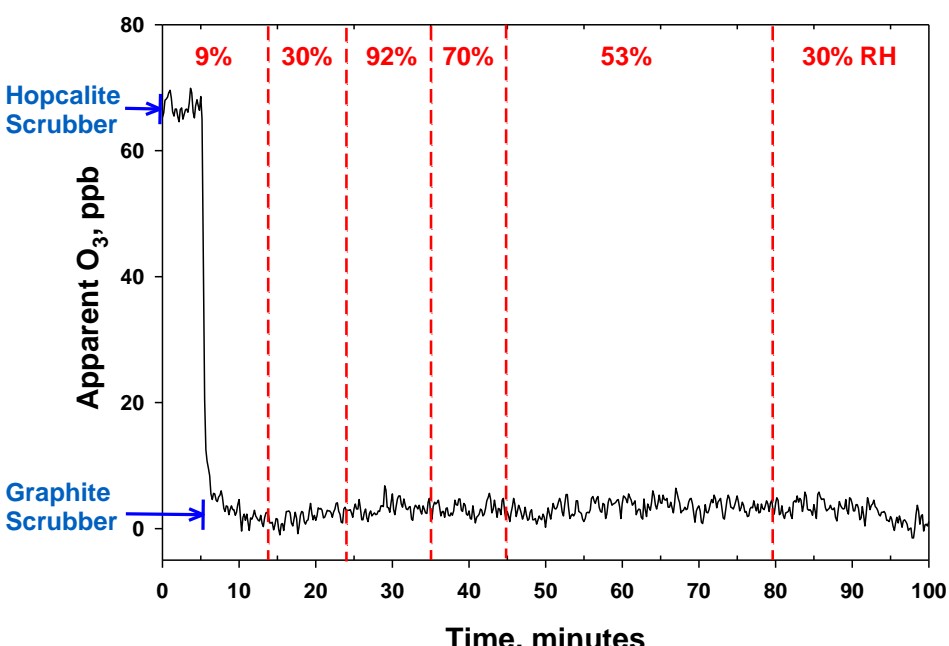

**Figure 9.** Apparent ozone observed upon addition of ~1.3 ppm of p-xylene to ozone-free air. Initially, the ozone analyzer was equipped with the standard Hopcalite scrubber (Apparent $O_3$ concentration= 67 ppb). The Hopcalite was replaced with the heated graphite scrubber at time = 5 minutes. Over the next 95 minutes, the relative humidity of the air was varied as designated by the red vertical dashed lines.

