# Peer review of "Use of a Heated Graphite Scrubber as a Means of Reducing Interferences in UV-Absorbance Measurements of Atmospheric Ozone"

_Atmospheric Measurement Techniques, 2017_

## Referee Comment (RC1) · Anonymous Referee #1 · 24 Mar 2017

The comment was uploaded in the form of a supplement:
http://www.atmos-meas-tech-discuss.net/amt-2017-3/amt-2017-3-RC1-supplement.pdf
* * *

---

## Referee Comment (RC2) · Anonymous Referee #2 · 25 Mar 2017

Review of Turnipseed et al., "Use of a Heated Graphite Scrubber as a Means of Reducing Interferences in UV-Absorbance Measurements of Atmospheric Ozone"

This paper describes a new alternative to conventional ozone scrubbing materials used within common UV-absorbance ozone analyzers. UV ozone analyzers, which are most commonly used for regulatory compliance monitoring, are susceptible to some degree of positive bias from interferents that both absorb at 253.7 nm and are scrubbed by the ozone scrubbing material for the Io measurement. With the tightening of the NAAQS ozone standard to 70 ppbv, it is prudent to works towards an improved ozone monitoring method that reduces the potential positive bias that could lead to false ozone non-attainment designations.

I view this current work as a first-step towards developing an improved UV-absorbance ozone monitor. This work demonstrates very well the much-improved performance of the new graphite scrubber in terms of reduced interferences from VOCs, water vapor, and mercury. The ability to omit a Nafion dryer is a definite benefit for ambient measurements. I have doubts regarding the real-world applicability of this method as it currently stands for compliance monitoring, primarily because dual-beam analyzers are overwhelmingly used for this application; however, I consider any incremental steps towards an improved method to be valuable in progressing the science. I also think that this method could have useful applications in laboratory studies, perhaps smog-chamber experiments, where VOC mixing ratios are typically much greater than ambient. Therefore, I do recommend publication of this work after addressing a few relatively minor concerns.

The primary complaint that I have with this manuscript is the somewhat misleading nature of the discussion regarding the potential positive biases with FEM ozone monitors. Although the authors do acknowledge that interferences in outdoor air normally cause only very small errors, "a few ppb at most", I think one who reads this paper without a background knowledge of ozone measurements or atmospheric science would draw the conclusion that most, or perhaps all, of the regulatory monitors are skewed high, and that a measurement error is resulting in non-attainment designations. Because EPA regulations, and associated non-attainment penalties, are an especially hot topic in today's political climate, the language used here needs to be cautious and make it clear that interferences of even a few ppb would be expected in only certain circumstances and in highly polluted environments. Ollison et al, 2013 reports positive biases of a few ppb from measurements conducted with the highly-industrialized Houston Ship Channel, a notoriously polluted location, though it's not discussed whether those few measurements from one location would be enough to designate the city as non-attainment. Other works cited in this manuscript present ambient measurement comparisons in Mexico City, a location with exceptionally high pollution relative to levels observed in the United States today. Although I know the authors are seeking to strengthen the motivation for this study, it is necessary to also acknowledge studies that have shown no discernible bias. Dunlea et al., 2006 is cited, but the "excellent agreement" they report between the UV monitor and the DOAS is not acknowledged. This manuscript should also cite Ryerson et al., 1998, in which no measurement bias was observed in concurrent O3 measurements by a chemiluminescence instrument and a UV monitor through 5 missions over 4 years, including within the Nashville urban plume. Parrish and Fehsenfeld, 2000, state "Even though significant evidence of interferences in the UV absorption technique has been reported, such interferences are not always observed, even in urban plumes."

The agreements in Figure 3a also suggest that the Hopcalite-srubbed 205 did not suffer any interferences in Boulder.

Along this same line, it is also essential to discuss quantitatively the levels of interferences one could reasonably expect from the compounds listed in Table 1 given typical ambient atmospheric mixing ratios. While I understand that laboratory studies and tests must use quite high VOC levels in order to generate the plots presented in Figure 7, it must be pointed out clearly that 1 ppm of xylene is not a realistic ambient atmospheric mixing ratio under normal circumstances. I'm stressing this strongly because AMT is an open-access journal, and one without an atmospheric science background likely is not aware of the normal atmospheric mixing ratios of these compounds. I would like to see two additional columns added to Table 1 that state the typical ambient mixing ratios of these compounds and then what that typical mixing ratio would equate to in "apparent" O3. I do appreciate that this is discussed in regards to mercury in ambient air on Lines 405-415. Pointing out the larger industrial emissions in the Houston Ship Channel, what compounds are enhanced there, and why this is a good example of a location where a positive bias has been shown to exist, would also be informative.

I have no doubts regarding the improved performance of the graphite scrubber, and I do believe that it could find valuable use in lab or smog-chamber studies where VOC mixing ratios are typically very high. However, going back to applicability to real-world monitoring, I wonder whether the uncertainty associated with the analyzer itself is even sufficient to discern any potential improvement by this scrubber. My personal experience with using the 202 Single-Beam, and that of others I have worked with who have had independent experiences, is that this monitor is generally very noisy and variable, making 1-min or less data essentially useless. The agreements shown in Figure 3 suggest that 5-min averaged data doesn't suffer to the same degree, but I still want to know what the measurement uncertainty is for the 5-min data. This is especially important to discuss given the statements on lines 490-494 that: "concentration levels of interfering VOCs were quite low in the Spicer et al (2010) study, ranging from 7.6 to 14 ppb and their measured apparent ozone mixing ratios were <15 ppb. At these levels, small signal drifts, or even the typical precision of ±1 to 2 ppb in the ozone analyzers impart significant measurement uncertainty." I would argue that 7.6 to 14 ppb is actually very high relative to typical atmospheric mixing ratios of these compounds; so, can any standard FEM analyzer even distinguish a potential bias from these VOCs within its measurement uncertainty (barring exceptional emissions events)? I understand that this work is about the performance of the scrubber and not the monitor, but my question is about whether this new scrubber actually improves the ozone measurement in practice in typical ambient measurements given the limitations of the monitor itself. I recommend addressing this issue somewhere in the manuscript and quantifying the uncertainty of the monitors used.

Additional comments:
- Line 75: The authors state that desorption at a later time would cause a measured negative absorbance. This is only true in ozone-free air (or perhaps ODEs?); in ambient air this would be a negative *bias*. Please clarify.
- I understand that there was not a mercury analyzer available to quantify what concentrations of mercury vapor were tested, but would it be possible to at least provide

an estimate of the range of mercury tested given the temperature, vapor pressure of mercury, and flow rates?

- In regards to the scrubber degradation discussion (pages 7 and 8), the laboratory degradation tests appear to have been conducted with relatively high O3 (150-250 ppb and then 300-700 ppb), and from this it was concluded that this scrubber isn't feasible for the dual-beam. Sampling ambient levels in Boulder, the scrubber lasted 38 days at 130° in the single-beam. So how long does the scrubber last at 130° sampling ambient air in the dual-beam? I would assume it must be better than the "overnight" time period deduced from the lab test at high O3 levels.
- Line 211: Define "adequate" in quantitative terms. What is the ozone destruction efficiency of the graphite and how does that compare to conventional scrubbers?
- Line 238" Define "high".
- Lines 244-246: How long is "overnight" exposure?
- Lines 246-247: Quantify "faster temporal decay." What is the decay rate?
- Line 248: Define "periodically." How often would one have to recharge the scrubber if operating under ambient conditions? Is it too often to make it worthwhile? Is it possible to just shut off the flow to the Io channel of a dual-beam monitor every so often for a few minutes to let it recharge? Or use two scrubber channels and switch between the two to continuously do an Io measurement in a dual-beam?
- Line 260-261: Is the competition also dependent upon the mixing ratio of O3 being sampled?
- Lines 427-430: Clarify that the "positive absorbance measurement" and "negative absorption" only apply to ozone-free air; or else change to the wording to *bias* rather than absorbance.
- Lines 473-474: Do you mean that error tends to *decrease* with volatility? It should be more difficult to desorb the less-volatile compounds, meaning the high-volatility compounds would have less error, if I'm understanding this correctly.
- Line 481: Since these selectivity ratios are calculated based upon an assumption and not directly measured, I suggest changing "were measured" to "were estimated."
- Lines 590-592: The concentrations of these species are a factor of what higher than typical ambient mixing ratios?
- Table 1: Besides the suggestion above, I suggest also adding naphthalene to the table since it is discussed in some of the cited references (e.g, the Spicer papers).
- Figure 3: Based on this agreement, is it fair to conclude that, at least for Boulder, no actual benefit is observable in ambient measurements using the new scrubber? For follow-up work, I suggest doing this comparison in a more polluted environment, like the Houston Ship Channel perhaps, where the benefit could be observed.

References:

Parrish, D.D. and Fehsenfeld, F.C.: Methods for Gas-Phase Measurements of Ozone: Ozone Precursors, Aerosol Precursors; Atmos. Environ. 34, 1921-1957, 2000.

Ryerson, T.B., et al.: Emissions lifetimes and ozone formation in power plant plumes. Journal of Geophysical Research 103, 22569-22583, 1998.

---

## Author Comment (AC1) · 1 May 2017

We would like to thank both referees for their thoughtful comments and suggestions and believe that this will improve the quality and content of our manuscript. We have attached a revised version of the manuscript along with our repsonses. Line numbers in our responses refer to the new revised manuscript. Referee comments are in blue – followed by our responses.

**Author Response to Referee #1.**

This paper describes the evaluation of a new scrubber to provide the zero measurement in UV absorption based ozone instruments. The motivation is that typical scrubbers remove interferences such as water and mercury. The removal of these species in the zero measurement leads to a measurement bias. The advantage of this new scrubber is that it does not remove these interferences and hence, it potentially provides a better measurement of ozone. The paper provides an extensive evaluation of the new graphite scrubber and compares it with the most common scrubbers used in commercial instruments. The presentation is clear and the results are significant and will be useful to the AMT readership. This paper is acceptable after minor changes to answer the comments below.

The authors would like to thank Referee #1 for their comments.

General: The paper is focused on using the scrubber for a 2B ozone monitor, which is appropriate, but it does neglect some more general applicability of this scrubber for lab use or for readers who use a different instrument. It would help to include information needed for other uses, too.

We agree with Referee #1 and appreciate the suggestion to consider further applications for our new heated graphite scrubber. In particular, researchers using smog chambers or chambers for aerosol studies often use high VOC concentrations to drive rapid photochemisty, but can lead to interferences with concurrent ozone measurements (note – Referee #2 also mentioned this application). We have added a sentence in the Introduction (beginning at line 61) that mentions this possible application along with corresponding references. We have also mentioned this application at the end of abstract and in the final conclusions.
We have also added a sentence (line 630) in the Conclusions section pointing out the feasibility of the graphite scrubber to replace existing scrubbers in conventional ozone monitors since the operating parameters (e.g., flow rate, pressure) are similar to existing scrubbers.

Line 135. Does the cleaning process affect the ozone removal efficiency? What about the results from the coarsely ground graphite? You should mention the results or remove the tease.
The cleaning process for the graphite did not affect the ozone removal efficiency – its primary effect appears to be removing impurities from the graphite (likely from the machining process) that have a tendency to remove or absorb several of the less volatile VOCs. We have clarified out statement to read (line 148): "Scrubbers consisting of coarsely ground EDM-AF5 graphite (~ 1-3 mm diameter particles) were also tested and exhibited similar properties for both ozone destruction and the reduction of interferences. However, we present results from the tube

scrubbers because their geometric surface area and residence times were more easily characterized."

 The products of the O3 + graphite reaction need better explanation and references. These references are really hard to find and do not provide the information that they are cited for. I could not access any from NASA and only one from University of Maryland (Tracz), and it was not at all useful to explain the products. Do you have any indication that acetone or acetaldehyde are produced? Did you look at the products with the GC-FID?

We thank Referee #1 for pointing this out.  We were only able to find a few direct studies of ozone and graphite and, as pointed out, they are more focused on surface characteristics than gas phase products.  We have moved two of the references earlier in the sentence such that they only cite evidence that a reaction exists (lline 202).  Razumovskii et al., (2007) do observe loss of graphite mass and assume this is due to loss of gaseous CO and $CO_2$.  CO and $CO_2$ have been identified as the major products of $O_3$ reacting with various forms of black carbon. Black carbon is graphitic in nature, but may not react exactly as pure graphite.  We have included a sentence to note product identifications in the $O_3$/black carbon reactions along with associated references (line 204).

We did not mean to imply that acetone and acetaldehyde are, in fact, produced from $O_3$/graphite reaction, and we did not attempt to look at products of the $O_3$/graphite reaction.  However, if hydrogen atoms are available – these seem likely carbon-containing candidates that would then be lost to the gas phase and could cause absorption of light in the optical cell.  Many different oxygen functional groups (carboxyl, hydroxyl, etc.) have been identified on the surface of black carbon after exposure to ozone (Akhter, et al., Appl Spectrosc.,45, 653-665, 1991; Chughtai et al., Aerosol Sci. Technol., 15, 112-126, 1991).  We have changed the text to read (line 206): "CO, $CO_2$, and other possible low volatility hydrocarbons which could be lost to the gas phase tend to show negligible absorption at 253.7 nm (Table 1) compared to ozone."

 What are the flow rates for these estimates?
We assume a typical flow rate for a single beam analyzer of 1 Lpm. This is now included in the text.

 Explain the reason for offset.
The exact cause of the small residual negative offset is unclear, although it is likely that there are still some residual organics that are slowly released from the hot graphite.  We have included this explanation in the text (line 222).

 Do you have any indication from the GC-FID whether the uptake of VOC on hopcalite and MnO is reactive? Do you know what comes off the scrubbers when they are subsequently heated?
We did not observe any other compounds eluting on the GC-FID.  This suggests a simple adsorptive uptake, but does not preclude chemical transformations since the product could (and may likely) be less volatile than the original VOC.  In the past we have used heated Hopcalite and still tend to see substantial uptake of VOCs (as evidenced by an apparent ozone absorption).

**Author Response to Referee #2**

This paper describes a new alternative to conventional ozone scrubbing materials used within common UV-absorbance ozone analyzers. UV ozone analyzers, which are most commonly used for regulatory compliance monitoring, are susceptible to some degree of positive bias from interferents that both absorb at 253.7 nm and are scrubbed by the ozone scrubbing material for the Io measurement. With the tightening of the NAAQS ozone standard to 70 ppbv, it is prudent to works towards an improved ozone monitoring method that reduces the potential positive bias that could lead to false ozone non-attainment designations.
I view this current work as a first-step towards developing an improved UV-absorbance ozone monitor. This work demonstrates very well the much-improved performance of the new graphite scrubber in terms of reduced interferences from VOCs, water vapor, and mercury. The ability to omit a Nafion dryer is a definite benefit for ambient measurements. I have doubts regarding the real-world applicability of this method as it currently stands for compliance monitoring, primarily because dual-beam analyzers are overwhelmingly used for this application; however, I consider any incremental steps towards an improved method to be valuable in progressing the science. I also think that this method could have useful applications in laboratory studies, perhaps smog-chamber experiments, where VOC mixing ratios are typically much greater than ambient. Therefore, I do recommend publication of this work after addressing a few relatively minor concerns.

We thanks Referee #2 for their time and effort in reviewing this manuscript.  We also agree with Referee #2 that this is an initial step towards improving UV-absorbance ozone measurements and that further studies are needed to substantiate the advantages of the heated graphite scrubber.  We do remark to this extent in the final paragraph of the Conclusions section. We do disagree that dual beam detectors are "overwhelmingly used" in compliance monitoring – the Teledyne-API 400 series monitors are single beam instruments and are used in a significant (and growing) fraction of compliance monitoring sites within the U.S.  However, we do now have evidence that the heated graphite scrubber can be used successfully in dual beam analyzers.  Details of this and specific changes to the manuscript are given below to address Referee #2's third Specific comment.  Finally, we have also included that the heated graphite scrubber may find utility in smog-chamber type experiments (this was also suggested by Referee #1) and included appropriate references in the 3rd paragraph of the Introduction (beginning on line 61) along with the discussion concerning indoor air. We thank both Referees for noting this possible application.

The primary complaint that I have with this manuscript is the somewhat misleading nature of the discussion regarding the potential positive biases with FEM ozone monitors. Although the authors do acknowledge that interferences in outdoor air normally cause only very small errors, "a few ppb at most", I think one who reads this paper without a background knowledge of ozone measurements or atmospheric science would draw the conclusion that most, or perhaps all, of the regulatory monitors are skewed high, and that a measurement error is resulting in non-attainment designations. Because EPA regulations, and associated non-attainment penalties, are an especially hot topic in today's political climate, the language used here needs to be cautious and make it clear that interferences of even a few ppb would be expected in only certain circumstances and in highly polluted environments. Ollison et al, 2013 reports positive biases of

a few ppb from measurements conducted with the highly-industrialized Houston Ship Channel, a notoriously polluted location, though it's not discussed whether those few measurements from one location would be enough to designate the city as non-attainment. Other works cited in this manuscript present ambient measurement comparisons in Mexico City, a location with exceptionally high pollution relative to levels observed in the United States today. Although I know the authors are seeking to strengthen the motivation for this study, it is necessary to also acknowledge studies that have shown no discernible bias. Dunlea et al., 2006 is cited, but the "excellent agreement" they report between the UV monitor and the DOAS is not acknowledged. This manuscript should also cite Ryerson et al., 1998, in which no measurement bias was observed in concurrent O3 measurements by a chemiluminescence instrument and a UV monitor through 5 missions over 4 years, including within the Nashville urban plume. Parrish and Fehsenfeld, 2000, state "Even though significant evidence of interferences in the UV absorption technique has been reported, such interferences are not always observed, even in urban plumes." The agreements in Figure 3a also suggest that the Hopcalite-srubbed 205 did not suffer any interferences in Boulder.

We fully understand Referee #2's concern that this work could be construed as a critical indictment of current regulatory ozone monitoring.  It is certainly not our intention to undermine the current network and we certainly realize that the advantages of the graphite scrubber are typically small in most instances with the exception of highly polluted areas.  We do maintain that small biases (few ppb) which are difficult to discern are possible and could pose problems in near-compliant areas.  We have re-written the second paragraph of the Introduction (starting at line 49) to (1) note the locations of the Ollison et al (2013) and Leston et al (2005) measurements and also noting that these locations typically have very high pollution levels and (2) including text noting the Ryerson et al (1998) and the Parrish and Fehsenfeld conclusions that no measurement bias was observed in their extensive work.  We did not include the "excellent agreement" reported by Dunlea et al (2006) in their Mexico City study – as they did actually observe differences between UV-absorbance and DOAS (+13 to – 18%).  They attributed this to incorrect calibration factors, but state that "interferences could not be completely ruled out."

Along this same line, it is also essential to discuss quantitatively the levels of interferences one could reasonably expect from the compounds listed in Table 1 given typical ambient atmospheric mixing ratios. While I understand that laboratory studies and tests must use quite high VOC levels in order to generate the plots presented in Figure 7, it must be pointed out clearly that 1 ppm of xylene is not a realistic ambient atmospheric mixing ratio under normal circumstances. I'm stressing this strongly because AMT is an open-access journal, and one without an atmospheric science background likely is not aware of the normal atmospheric mixing ratios of these compounds. I would like to see two additional columns added to Table 1 that state the typical ambient mixing ratios of these compounds and then what that typical mixing ratio would equate to in "apparent" O3. I do appreciate that this is discussed in regards to mercury in ambient air on Lines 405-415. Pointing out the larger industrial emissions in the Houston Ship Channel, what compounds are enhanced there, and why this is a good example of a location where a positive bias has been shown to exist, would also be informative.

We have added a column in Table 1 giving typical ranges of many of the compounds shown. In adding this column – we have removed the column for boiling point (which is somewhat

redundant to vapor pressure) for simplification.  Most of these concentrations were found in Atmospheric Chemistry textbooks or papers showing comparison between multiple studies to give an overall general concentration range. We have not added a column for the expected "apparent O3" produced, but have included in the table footnotes: "To estimate the maximum response that a particular interferent could give in an ozone monitor, divide the Typical Mixing Ratio by the Selectivity Ratio, S"  Hopefully this will clarify what levels of bias could normally be expected.

I have no doubts regarding the improved performance of the graphite scrubber, and I do believe that it could find valuable use in lab or smog-chamber studies where VOC mixing ratios are typically very high. However, going back to applicability to real-world monitoring, I wonder whether the uncertainty associated with the analyzer itself is even sufficient to discern any potential improvement by this scrubber. My personal experience with using the 202 Single-Beam, and that of others I have worked with who have had independent experiences, is that this monitor is generally very noisy and variable, making 1-min or less data essentially useless. The agreements shown in Figure 3 suggest that 5-min averaged data doesn't suffer to the same degree, but I still want to know what the measurement uncertainty is for the 5-min data. This is especially important to discuss given the statements on lines 490-494 that: "concentration levels of interfering VOCs were quite low in the Spicer et al (2010) study, ranging from 7.6 to 14 ppb and their measured apparent ozone mixing ratios were <15 ppb. At these levels, small signal drifts, or even the typical precision of ±1 to 2 ppb in the ozone analyzers impart significant measurement uncertainty." I would argue that 7.6 to 14 ppb is actually very high relative to typical atmospheric mixing ratios of these compounds; so, can any standard FEM analyzer even distinguish a potential bias from these VOCs within its measurement uncertainty (barring exceptional emissions events)? I understand that this work is about the performance of the scrubber and not the monitor, but my question is about whether this new scrubber actually improves the ozone measurement in practice in typical ambient measurements given the limitations of the monitor itself. I recommend addressing this issue somewhere in the manuscript and quantifying the uncertainty of the monitors used.

Conventional UV-absorbance ozone monitors (including 2B analyzers) have stated measurement precisions of ≤ 1.5 ppb (in fact, analyzers from ThermoFisher and Teledyne-API only tend to output running averages which result in lower apparent noise). Most compliance monitoring is done on an hourly averaged basis which reduces the measurement standard error even further – thus a bias of just a few ppb would be measurable. We included a statement in the Introduction (line 52) stating that a small (few ppb) bias would typically be measurable considering the hourly time average. The Model 202 and 205 analyzers used in this study were operating at their given specifications (see http://www.twobtech.com/docs/tech_notes/TN001.pdf). We have added a statement in the Experimental stating this (line 136).

In reference to the concentrations used the Spicer et al (2010) study, we agree that the concentrations they used are high relative to atmospheric levels.  We merely wanted to point out that by working at such low concentrations their reported response ratios will include a significant amount of error (e.g., a precision of 1.5 ppb immediately implies a minimum error of

±10% for a measured apparent ozone of 15 ppb). Spicer et al (2010) give no details how long measurements were averaged nor how VOC concentrations were confirmed.

Additional comments:
- Line 75: The authors state that desorption at a later time would cause a measured negative absorbance. This is only true in ozone-free air (or perhaps ODEs?); in ambient air this would be a negative *bias*. Please clarify.

Referee #2 is correct – we have changed the text to read: " …causing a negative absorbance (value of $I_o$ is less than I in ozone-free air), thus imparting a negative bias to the overall measurement."

- I understand that there was not a mercury analyzer available to quantify what concentrations of mercury vapor were tested, but would it be possible to at least provide an estimate of the range of mercury tested given the temperature, vapor pressure of mercury, and flow rates?

We have added a sentence in the Experimental section (line 175): "From bath temperature, flow rates and the mercury vapor pressure, estimated mercury vapor concentrations of 0.3 to 30 ppb were produced." These are larger concentrations than estimations based on Figure 6 and assuming that mercury absorbs ~ 1350 times more strongly than ozone (line 369). We have found that this is due to under-saturation of mercury vapor in the "sweep" flow of our diffusion tube.

- In regards to the scrubber degradation discussion (pages 7 and 8), the laboratory degradation tests appear to have been conducted with relatively high O3 (150-250 ppb and then 300-700 ppb), and from this it was concluded that this scrubber isn't feasible for the dual-beam. Sampling ambient levels in Boulder, the scrubber lasted 38 days at 130° in the single-beam. So how long does the scrubber last at 130° sampling ambient air in the dual-beam? I would assume it must be better than the "overnight" time period deduced from the lab test at high O3 levels.

Referee #2 is correct – as mentioned in our response to one of the general comments above, we have just recently gone back and investigated using the graphite scrubber in a 2B dual-beam Model 211 ozone monitor and have found that operating the scrubber at the higher 130°C setting also allows for quantitative ozone destruction in dual beam instruments. Currently we have exposed a single graphite scrubber (at 130°C) to: (1) ~ 10 ppm-hr of ozone in the laboratory (at concentrations of 200 and 400 ppb) and then an additional 10 ppm-hr of ozone over 20 days of ambient measurements. Comparisons of the ambient measurements with our standard FEM Model 205 are in excellent agreement and we have observed no loss in the ozone destruction efficiency thus far (measurements are ongoing). We have included a sentence in the last paragraph of Section 3.2 (line 310) stating: "Subsequent measurements using a dual beam Model 211 equipped with the heated graphite scrubber operating at 130°C indicate that there is also no loss in the ozone removal efficiency after exposure to 20 ppm-hr of ozone." as well as in the Conclusions section (line 595) stating: "These conditions appear to be valid for both single-beam and dual-beam monitors." We have also removed language suggesting that the graphite scrubber should not currently be used in dual-beam monitors.

- Line 211: Define "adequate" in quantitative terms. What is the ozone destruction efficiency of the graphite and how does that compare to conventional scrubbers?

We have adding "(> 98%)" for quantifying the ozone destruction efficiency which is comparable to other conventional scrubbers.

- Line 238" Define "high".

Razumovskii et al report 2.2 x 10$^{-4}$ mol/L which is ~ 0.5% at 1 atm and 298K. We have included this in the text.

- Lines 244-246: How long is "overnight" exposure?

18 hours – we have added this to the text.

- Lines 246-247: Quantify "faster temporal decay." What is the decay rate?

O3 decayed to 85% removal efficiency over 2 to 8 hours at the higher concentrations. We have added the approximate time (in hours) that it took before leveling off at 85% ozone removal efficiency at the higher ozone concentrations (line 264).

- Line 248: Define "periodically." How often would one have to recharge the scrubber if operating under ambient conditions? Is it too often to make it worthwhile? Is it possible to just shut off the flow to the Io channel of a dual-beam monitor every so often for a few minutes to let it recharge? Or use two scrubber channels and switch between the two to continuously do an Io measurement in a dual-beam?

The time frame between recharging the scrubber would depend upon the ozone exposure, but a conservative estimate would be every 1 to 2 days. As the recharging could be done at night when ozone levels are low anyway, this would not be too detrimental to routine monitoring. The other suggestions posed by Referee #2 (allowing a "rest period" for the Io or using two scrubbers) are both viable options and we have considered both of those. However, as mentioned in our response above – further studies suggest that we can use the graphite scrubber in a dual beam analyzer just by proper choice of the temperature.

- Line 260-261: Is the competition also dependent upon the mixing ratio of O3 being sampled?

Our observations suggest this and we have added that to the sentence in line 278.

- Lines 427-430: Clarify that the "positive absorbance measurement" and "negative absorption" only apply to ozone-free air; or else change to the wording to *bias* rather than absorbance.

Referee #2 is correct – VOCs would impart either a positive or negative bias to the measurements. We have changed the text accordingly (line 446).

- Lines 473-474: Do you mean that error tends to *decrease* with volatility? It should be more difficult to desorb the less-volatile compounds, meaning the high-volatility compounds would have less error, if I'm understanding this correctly.

Referee #2 is correct – our mistake – the error does decrease with volatility.

- Line 481: Since these selectivity ratios are calculated based upon an assumption and not directly measured, I suggest changing "were measured" to "were estimated."

Wording changed as suggested.

- Lines 590-592: The concentrations of these species are a factor of what higher than typical ambient mixing ratios?

At line 609, we have added the sentence: "These are nearly three or more times greater than typical ambient concentrations for any of these species (Table 1)."

- Table 1: Besides the suggestion above, I suggest also adding naphthalene to the table since it is discussed in some of the cited references (e.g, the Spicer papers).

We have added naphthalene as suggested. We have removed a few species which are only observed in specific circumstances (e.g., acetonitrile which is typically a tracer for biomass burning).

- Figure 3: Based on this agreement, is it fair to conclude that, at least for Boulder, no actual benefit is observable in ambient measurements using the new scrubber? For follow-up work, I suggest doing this comparison in a more polluted environment, like the Houston Ship Channel perhaps, where the benefit could be observed.

We agree with Referee #2 that this scrubber needs to be tested in a more polluted environment to see its benefits and we are making efforts to undertake that study. We also did not expect to see much (if any) benefit from this scrubber in our Boulder measurements which is a fairly clean location. We do state that we expected that interferences would be < 3 ppb during those measurements.

References:
Parrish, D.D. and Fehsenfeld, F.C.: Methods for Gas-Phase Measurements of Ozone: Ozone Precursors, Aerosol Precursors; Atmos. Environ. 34, 1921-1957, 2000.
Ryerson, T.B., et al.: Emissions lifetimes and ozone formation in power plant plumes. Journal of Geophysical Research 103, 22569-22583, 1998.